

# The On-Orbit Performance of the Orbiting Carbon Observatory-2 (OCO-2) Instrument and its Radiometrically Calibrated Products

David Crisp[1], Harold R. Pollock[1], Robert Rosenberg[1], Lars Chapsky[1], Richard A. M. Lee[1], Fabiano A. Oyafuso[1], Christian Frankenberg[1,2], Christopher W. O'Dell[3], Carol J. Bruegge[1], Gary B. Doran[1],

Annmarie Eldering[1], Brendan M. Fisher[1], Dejian Fu[1], Michael R. Gunson[1], Lukas Mandrake[1], Gregory B. Osterman[1], Florian M. Schwandner[1,4], Kang Sun[5], Tommy E. Taylor[2], Paul O. Wennberg[2], and Debra Wunch[2,6]

[1] Jet Propulsion Laboratory/California Institute of Technology, Pasadena, CA, USA 91109-8099
[2] California Institute of Technology, Pasadena, CA, USA

[3] Colorado State University, Fort Collins, CO, USA
[4] Joint Institute for Regional Earth System Science and Engineering, University of California Los Angeles, Los Angeles CA, USA
[5] Harvard-Smithsonian Center for Astrophysics, Cambridge, MA, USA
[6] University of Toronto, Toronto, Canada

*Correspondence to*: David Crisp (David.Crisp@jpl.nasa.gov)

**Abstract.** The Orbiting Carbon Observatory-2 (OCO-2) carries and points a three-channel imaging grating spectrometer designed to collect high-resolution, co-boresighted spectra of reflected sunlight within the molecular oxygen ($O_2$) A-band at 0.765 microns and the carbon dioxide ($CO_2$) bands at 1.61 and 2.06 microns. These measurements are calibrated and then combined into soundings that are analyzed to retrieve spatially resolved estimates of the column-averaged $CO_2$ dry air mole

fraction, $X_{CO2}$. Variations of $X_{CO2}$ in space and time are then analyzed in the context of the atmospheric transport to quantify surface sources and sinks of $CO_2$. This is particularly challenging remote sensing observations because the all but the largest emission sources and natural absorbers produce only small ($< 0.25\%$) changes in the background $X_{CO2}$ field. High measurement precision is therefore essential to resolve these small variations and high accuracy is needed because small biases in the retrieved $X_{CO2}$ distribution could be misinterpreted as evidence for $CO_2$ fluxes.

To meet its demanding measurement requirements, each OCO-2 spectrometer channel collects 24 spectra per second across a narrow ($< 10$ km) swath as the observatory flies over the sunlit hemisphere, yielding almost one million soundings each day. On monthly time scales, between 7 and 12% of these soundings are sufficiently cloud free to yield full-column estimates of $X_{CO2}$. Each of these soundings has an unprecedented combination of spatial resolution ($< 3$ km$^2$ / sounding), spectral resolving power ($\lambda/\Delta\lambda > 17{,}000$), dynamic range ($\sim 10^4$), and sensitivity (continuum signal-to-noise ratio $> 400$).

The OCO-2 instrument performance was extensively characterized and calibrated prior to launch. In general, the instrument has performed as expected during its first 18 months in orbit. However, ongoing calibration and science analysis activities have revealed a number of subtle radiometric and spectroscopic challenges that affect the yield and quality of the OCO-2 data products. These issues include increased numbers of bad pixels, transient artifacts introduced by cosmic rays, radiance discontinuities for spatially non-uniform scenes, a misunderstanding of the instrument polarization orientation, and time-





dependent changes in the throughput of the oxygen A-band channel. Here, we describe the OCO-2 instrument, its data products, and its on-orbit performance. We then summarize calibration challenges encountered during its first 18 months in orbit and the methods used to mitigate their impact on the calibrated radiance spectra distributed to the science community.

Copyright statement

The JPL author's copyright for this publication is held by the California Institute of Technology. Government sponsorship acknowledged.

## 1 Introduction

The Orbiting Carbon Observatory–2 (OCO-2) is the first NASA satellite designed to measure the column-averaged carbon dioxide ($CO_2$) dry air mole fraction, $X_{CO2}$, with the accuracy, resolution, and coverage needed to identify and characterize $CO_2$ sources and sinks on regional scales over the globe. Surface weighted $X_{CO2}$ estimates can be retrieved from high resolution spectroscopic observations of reflected sunlight in near infrared $CO_2$ and molecular oxygen ($O_2$) bands (Rayner and O'Brien, 2001; Crisp et al. 2004; 2008; 2015; O'Dell et al. 2012). This is a particularly challenging space based remote sensing
measurement, because even the largest $CO_2$ sources and sinks produce changes in the background $X_{CO2}$ distribution that rarely exceed 0.25% (e.g. 1 part per million (ppm) out of the ambient 400 ppm background, http://cdiac.ornl.gov/ ) on spatial scales ranging from that of a satellite footprint (a few square km) to continental scales (Miller et al. 2007). These small variations in $X_{CO2}$ typically introduce changes the line core-to-continuum intensity ratio for $CO_2$ absorption lines that are smaller than 0.1% in reflected solar spectra, even at spectral resolving powers as high as $\lambda/\Delta\lambda = 20,000$.

To record these small changes in the reflected solar spectrum, OCO-2 carries and points a 3-channel imaging grating spectrometer that collects high resolution spectra of reflected sunlight in the 0.765 micron (µm) $O_2$ A-band and in the 1.61 and 2.06 µm $CO_2$ bands with an unprecedented combination spatial resolution (< 3 km$^2$ / sounding), spectral resolving power ($\lambda/\Delta\lambda$ > 17,000), sensitivity, and dynamic range (~$10^4$). Each spectrometer collects 24 spectra per second across a narrow (< 10 km) swath as the observatory flies over the sunlit hemisphere. Coincident measurements from the three spectral channels are
combined into "soundings" that are analyzed with a "full-physics" retrieval algorithm to yield estimates of $X_{CO2}$ and other geophysical quantities (Bösch et al. 2006; 2011; O'Dell et al. 2012; Crisp et al. 2012; Eldering et al. 2016).
OCO-2 was successfully launched from Vandenberg Air Force Base in California on 2 July 2014. After completing a series of spacecraft check-out activities and orbit raising maneuvers, it was inserted at the front of the 705-km Afternoon Constellation (A-Train; L'Ecuyer and Jiang, 2010) on 3 August 2014. The optical bench and focal planes of the three-channel imaging
grating spectrometer were then cooled to their operating temperatures (near -152.4 °C and -6.4 °C, respectively) and a series of calibration and validation activities was initiated. Since 6 September of 2014, the OCO-2 instrument has been routinely returning almost one million soundings each day over the sunlit hemisphere. These spectra have single-sample continuum


signal-to-noise ratios (SNR) exceeding 400, even over dark ocean surfaces at solar zenith angles as high as 80 degrees. On monthly time scales, 7 to 12 % of these soundings are sufficiently cloud free to yield full-column estimates of $X_{CO2}$. Nadir soundings over land yield $X_{CO2}$ estimates with single-sounding random errors that increase from 0.5 ppm to 1 ppm between the sub-solar latitude and solar zenith angles near 60 degrees. Observations collected near the apparent glint spot, where light

is specularly reflected by the Earth's surface, yield $X_{CO2}$ estimates over the ocean with single sounding random errors near 0.5 ppm at solar zenith angles as high as 70 degrees (Eldering et al. 2016).

Here we describe the instrument performance over the first 18 months of the OCO-2 mission. Sections 2 and 3 provide a brief overview of the instrument design and on-orbit calibration system. Section 4 describes the data products delivered to the science community. Section 5 summarizes the on-orbit radiometric, spectroscopic, and geometric performance of the

instrument. Section 6 describes the calibration challenges encountered in orbit and the changes in the calibration and data processing approach implemented to address these issues and to increase the yield and quality of the data products.

## 2 The OCO-2 Instrument

The OCO-2 spacecraft carries and points a single instrument that incorporates three, co-boresighted, long-slit, imaging grating spectrometers optimized for observing the molecular oxygen ($O_2$) A-band at 0.765 μm and the "weak" and "strong" $CO_2$ bands

at 1.61 and 2.06 μm (Crisp et al. 2004; Haring et al. 2004; Crisp et al. 2008; Pollock et al. 2010). These three channels are designated ABO2, WCO2, and SCO2, respectively. All three spectrometers use similar optical designs and are integrated into a common optical bench assembly (OBA) to improve system rigidity and thermal stability. They share a common, 200 mm focal length, F/1.8 Cassegrain telescope.

The optical path in each spectral channel is shown in Figure 1, along with examples of recorded spectra. Light entering the

telescope is focused at a field stop and then re-collimated before entering a relay optics assembly. The relay optics use dichroic beam splitters to direct the light to the SCO2, WCO2, and ABO2 channels, in that order. The light then traverses a narrowband pre-disperser filter designed to transmit wavelengths within ± 1% of the central wavelength of the channel and reject the rest. The light is then refocused onto each spectrometer slit by a reverse Newtonian telescope (Haring et al. 2004). Each slit is about 3 mm long and 28 μm wide. These long, narrow slits are aligned to produce co-boresighted fields of view that are about

0.0145 radians (0.83°) wide at the focal plane. Linear polarizers are installed in front of each spectrometer slit to pass only that component of light whose polarization is most efficiently diffracted by the spectrometer grating.

Once the light enters spectrometer slit, it is collimated by a 2-element refractive collimator and then dispersed by a gold-coated, reflective, planar, holographic diffraction grating that works in 1st order. The dispersed light is then focused by a 2-element refractive camera lens assembly onto a 2-dimensional focal plane array (FPA). A second, narrowband filter, which is cooled

to approximately -93 °C , has been installed just above each FPA to reject thermal emission from the instrument at wavelengths > 2% away from the central wavelength of the channel.



The spectrometer optics produce a spatially-resolved, 2-dimensional image of a spectrum on a 1024 by 1024 pixel FPA with 18 µm by 18 µm pixels (Figure 2a). All three of the FPAs used by OCO-2 are HAWAII-1RG™ detectors that were manufactured by Teledyne Scientific and Imaging, LLC for the original Orbiting Carbon Observatory (OCO) mission (Pollock et al., 2010). The FPA in the ABO2 channel uses a silicon (HyViSi™) photodetector array, while the $CO_2$ channels use mercury

5    cadmium telluride (HgCdTe) photodetector arrays with a standard 2.5 µm long-wavelength cut-off.

The grating disperses the spectrum the direction perpendicular to the long axis of the slit, illuminating 1016 of the 1024 FPA columns. The four columns at each edge of the FPA are masked and returned as reference pixels. The full width at half-maximum (FWHM) of the slit image on the FPA (also called the Instrument Line Shape, or ILS) is sampled by 2 to 3.5 pixels in the direction of dispersion. The spectral range, spectral resolving power, number of spectral samples across the ILS FWHM

10   (spectral sampling), and dynamic range for each spectrometer are summarized in Table 1 (Frankenberg et al., 2014; Rosenberg et al., 2016; Lee et al., 2016).  The spectral sampling and resolving power vary across the wavelength range of each channel due to the anamorphic magnification introduced by the grating (Pollock et al. 2010).

**Table 1. Instrument Properties.**

| Channel | ABO2 | WCO2 | SCO2 |
|---|---|---|---|
| Spectral Range (µm) | 0.7576 to 0.7726 | 1.5906 to 1.6218 | 2.0431 to 2.0834 |
| Resolving Power | 17500 – 18500 | 19100 – 20500 | 19700 – 19900 |
| Spectral sampling | 2.6 - 3.5 | 2.3 - 3.2 | 2.2 - 3.2 |
| Dynamic Range (photons m$^{-2}$ µm$^{-1}$ sr$^{-1}$ s$^{-1}$) (Minimum to maximum measureable signal) | $7.5 \times 10^{16}$ to $7.0 \times 10^{20}$ | $2.15 \times 10^{16}$ to $2.45 \times 10^{20}$ | $2.15 \times 10^{16}$ to $1.25 \times 10^{20}$ |

The ABO2, WCO2, and SCO2 FPAs are cooled to operating temperatures near -157, -153, and -152 °C, respectively by a pulse tube cryocooler that is connected the FPAs by thermal straps. The cryocooler rejects its heat though heat pipes that are connected to an external radiator. The temperature of the OBA is maintained near -6.4 °C by a thermal shroud (Haring et al., 2008). The shroud rejects its heat through heat pipes that are connected to a passive external radiator. Because the FPAs and

20   OBA are at very different temperatures, they are linked by a low-conductivity, cryogenic subsystem (Lamborn, 2008; Na-Nakornpanom et al. 2015).

For routine science operations, a 220 (spatial rows) by 1016 (spectral columns) pixel window on each FPA is continuously scanned using a "rolling readout" method for recording and resetting the FPAs (Haring et al. 2004). This readout approach precludes the need for a physical shutter and eliminates spatial gaps between the exposures. The 0.83° field of view illuminated





by the 3-mm long spectrometer slits is sampled by about 190 pixels in the dimension orthogonal to the direction of dispersion (FPA rows). Science measurements are restricted to the central 160 of these 190 pixels, which are fully illuminated by the slits. The active 220 by 1016-pixel region on each FPA can be read out pixel-by-pixel, but this readout mode is not practical for routine science observations because it takes 9.333 seconds for the instrument signal chain to process this much data. The

response of each pixel in the 220 by 1016 active area is therefore only recorded for FPA characterization and spectrometer calibration, and not for routine science data collection.

To reduce the FPA readout time and downlink data volume, the 160 rows that are fully illuminated by the slit are divided into 8 discrete "footprints", each consisting of 19 to 21 adjacent spatial pixels along an FPA column (the "Spatial Direction" in Figure 2a). The pixels within each footprint are summed together on-board by the instrument processor. In this "summed

mode," the active area of the FPAs can be read out at 3 Hz. The 8-footprint "image" of the spectrum that is recorded for each 0.333 second exposure is called a "frame." The ~20-pixel sum corresponding to a given column on the FPA is defined as a "spectral sample." The along-slit angular field of view of each spatially-averaged spectral sample is about 1.8 milliradians (mrad), or ~0.1°. The spatial field of view recorded by the 1016 spectral samples in a science spectrum is defined as a "summed footprint." In addition to reducing the data volume and readout time, this on-board summing reduces the impact of single-pixel

read noise (which decreases with the square root of the number of pixels included in the sum), increasing spectral sample SNR in low illumination conditions.

The angular width of the narrow dimension of the slit is only 0.14 mrad, but the focus of the entrance telescope was intentionally built with spherical aberrations to increase the effective full width at half maximum of each slit to ~0.5 mrad. This blurring was introduced to simplify the boresight alignment of the 3 spectrometer slits and to reduce the impact of bad

pixels incorporated into spectral samples on the FPAs. Over the 0.333 second integration time for each frame, the spacecraft moves about 2.25 km down-track, yielding 8 summed footprints with dimensions of ~2.25 km by 1.3 km at nadir when the instrument is operated in "push-broom" fashion with the slit oriented orthogonal to the ground track. Because the rolling readout scans across the 220 active rows sequentially over the 0.333-second exposure while the spacecraft is moving, the surface footprints are shaped like parallelograms rather than squares, even when operated in push-broom fashion (Figure 2b).

In addition to the 8, spatially-summed, 1016-element spectra in each frame, each spectrometer returns up to 20 FPA columns (colors) from each FPA without any on-board spatial summing. These "color slices" sample the full, along-slit spatial dimension at single-pixel resolution. Each color slice records a 220-pixel wide region of the FPA, which includes the full field illuminated by the slit (~190 pixels) as well as a few pixels beyond the ends of the slit that are not directly illuminated. These color slices are used to detect spatial variability within each of the spatially summed spectral samples. Color slice pixels outside

the 190-pixel wide region directly illuminated by the slit can also be used to quantify the thermal radiation and scattered light within the instrument. The FPA columns (wavelengths) for the 20 color slices are specified by commands from the ground.





## 3 The On-Board Calibration System

An onboard calibrator (OBC) has been integrated into the telescope baffle assembly, which is mounted on an external panel of the spacecraft bus (Figure 3). This system consists of a calibration "paddle" that carries an aperture cover (lens cap) on one end and a transmission diffuser on the other (Haring et al. 2008). The cover is placed over the telescope aperture to protect the

instrument from external contamination during launch and orbit maintenance activities. It is also closed to acquire "dark frames" that are used to monitor the zero-level offset of the FPAs. The back side of the cover has a diffusively reflecting gold surface that can be illuminated by one of 3 tungsten halogen lamps installed in the baffle assembly. These lamp "flat field" images are used to monitor changes in the relative gain of the individual pixels and spectral samples.

The calibration paddle is rotated 180° from the closed position to place the transmission diffuser in front of the telescope

aperture to acquire observations of the Sun. The diffuser is an all-reflective design that consists of a pair of plates separated by a cavity that is a few millimeters wide. The surfaces facing the target and the telescope aperture include arrays of pinholes that are not aligned. The patterned inner surfaces of the two plates are coated by textured gold surfaces. Sunlight that enters the pin holes on the target (Sun) facing side of the diffuser is reflected multiple times by the roughened gold surfaces between the plates before reaching a pinhole on the telescope-facing side of the diffuser, where it enters the optical path to the spectrometers.

The number and size of the pinholes were designed to yield diffuse intensities similar to those of a moderately dark ($< 5\%$) reflecting Lambertian surface when the telescope is pointed at the Sun.

The solar diffuser is used to acquire routine observations of the Sun just after the spacecraft crosses the northern terminator on all orbits except those that include downlinks. These measurements are used to monitor the absolute radiometric response of the instrument. The solar diffuser is also used periodically to acquire spectra of the Sun over the entire illuminated part of an

orbit. These observations sample the full range of solar relative velocities ($\pm$ 7 km/sec) and associated Doppler shifts observed over the illuminated hemisphere. The Doppler-shifted spectra collected over each Solar Doppler orbit can be combined to produce a massively over-sampled solar spectrum. These data are used to monitor variations in the instrument line shape (ILS) and have also been used to refine our understanding of the top-of-atmosphere solar spectrum for the wavelength ranges sampled by OCO-2. The calibration paddle is rotated 90° from either the closed or diffuser positions to open the telescope aperture for

routine science or lunar calibration observations. Lunar calibration observations were originally planned to monitor the instrument's geometric calibration. They are also being used to monitor the degradation of the calibration system's lamps and solar and lamp diffusers.

## 4 OCO-2 Radiance Data Products

The OCO-2 mission produces a "Level 1B" (L1B) product consisting of full orbits or fractions of orbits of calibrated and

geolocated spectral radiances from the ABO2, WCO2, and SCO2 channels. These products are screened for optically thick clouds and then analyzed with a full-physics remote sensing retrieval algorithm to yield estimates of $X_{CO2}$ and other geophysical parameters that constitute the OCO-2 Level 2 (L2) data products. For routine science operations, the spacecraft



points the instrument's boresight at the local nadir or in the vicinity of the "glint spot," where sunlight is specularly reflected from the Earth's surface. In a typical nadir-observation orbit, the instrument collects data for about 46.5 minutes and each spectrometer collects about 8360 frames (66880 spatially-resolved spectra) at solar zenith angles as high as 85°. For a typical glint-observation orbit, the instrument records science data for about 49.3 minutes and each spectrometer collects about 8880

frames (71000 spatially-resolved spectra) at solar zenith angles as high as 84°. Each L1B file or "granule" included in this product contains a record for each spectral sample and footprint in each channel, for each frame the instrument collects while viewing Earth during a single spacecraft orbit. All soundings that pass a series of quality criteria (e.g. spectra, geolocation, and housekeeping data available in all 3 channels, science aperture open) are included in the L1B product.

The OCO-2 L1B products are delivered to the NASA Goddard Earth Science Data and Information Services Center (GES

DISC) for distribution and archiving (http://disc.sci.gsfc.nasa.gov/OCO-2). The first L1B products were delivered to the GES DISC in December 2014. During the summer of 2015, all L1B and L2 data collected since routine operations began on 6 September 2014 were reprocessed using Version 7 (V7) of the OCO-2 algorithm. Both "forward" and "retrospective" versions of the Version 7 L1B and L2 products are routinely being delivered to the GES DISC. The "forward" product, designated V7, uses calibration information based on extrapolations of calibration data that were collected in the recent past. These products

are usually delivered to the GES-DISC within one week of acquisition. These V7 "forward" products are generally reliable except when the instrument experiences a significant change in its thermal environment that was outside of the training range used to derive the extrapolated calibration data. The "retrospective" L1B products, designated V7r, use interpolated calibration data and are therefore expected to be more reliable than the forward products. The V7r products are typically delivered to the GES-DISC in one-month blocks, starting about 5-6 weeks after acquisition. The OCO-2 team recommends that the V7r product

be used in applications where high accuracy is needed. The V7 and V7r products are described in greater detail in the OCO-2 Data Product User's Guide and in the L1B Algorithm Theoretical Basis Document (ATBD), which are posted along with the products at the GES DISC (http://disc.sci.gsfc.nasa.gov/OCO-2/documentation/oco-2-v7).

## 5 On-Orbit Performance

In general, the OCO-2 instrument has performed as expected, yielding spectra of reflected sunlight with an unprecedented

combination of spatial resolution, spectral resolution, dynamic range, and signal to noise ratio (SNR). To ensure that the instrument calibration remains stable between calibration opportunities, the OBA temperatures have been controlled to stay within ± 0.2 °C of their nominal set points and the FPA temperatures are controlled to stay within ± 0.4 °C of their nominal set points during routine operations (Figure 4). Instrument decontamination (decon) cycles are performed periodically to remove ice that accumulates on the thermal straps that connect the cryocooler's cold head to the FPAs. In a typical decon

cycle, the OBA and FPAs are warmed to near room temperature (either 12 or 28 °C) for a few days, and then cooled back down to their operating temperatures. No science observations can be collected during these ~1 week periods. Fortunately, the



frequency of decon cycles has decreased over time as the instrument has lost water to space. Key features of the on-orbit radiometric, spectroscopic, and geometric performance are summarized in the following sub-sections.

## 5.1     Radiometric Performance

The radiometric performance of the OCO-2 spectrometers was established prior to launch through an extensive series of measurements of a calibrated integrating sphere (Rosenberg et al. 2016). These and other pre-launch tests (Frankenberg et al., 2014) indicated that the SNR of all three channels substantially exceeded the requirements. The median SNR of individual spectral samples in each channel recorded during a typical nadir orbit are shown in Figure 5a. In this example, the intensities in the ABO2, WCO2, and SCO2 channels spanned ~40%, 80, and 52% of their maximum measurable signals, respectively

(see Table 1 and Rosenberg et al., 2016). Maps of the continuum SNR for cloud-free soundings used to retrieve $X_{CO2}$ estimates in the V7r L2 product for April 2015 are shown in Figure 5b. Here, the mean continuum SNR values for each channel have been averaged into 2° by 2° bins. With these high SNR values, the range of solar zenith angles or latitudes where $X_{CO2}$ can be retrieved is restricted more by clouds and other limitations in the OCO-2 L2 algorithm than by low instrument sensitivity.

The largest SNR values are usually recorded over the Sahara and Australian deserts, but occasionally, even brighter values are

recorded in glint observations over the ocean at low latitudes. These bright glint scenes sometimes yield continuum intensities substantially exceeding the maximum measureable signal in one or more of the channels, producing saturated spectra. Saturated channels typically recover within one or two exposures, as expected, with no measurable residual image (see discussion in Rosenberg et al., 2016).

Since launch, the radiometric performance has been routinely monitored by deploying the solar diffuser in front of the

instrument aperture to collect observations of the Sun, or by placing the lamp diffuser in front of the aperture to collect both dark (lamp off) and flat field (lamp on) exposures. Dark, solar and lamp diffuser observations are collected in "summed" and or "single pixel" mode on alternate orbits that are not used for downlink or other maintenance activities. These routine radiometric calibration activities are augmented by observations of the Moon, collected for near full and gibbous phases each month. In addition, observations of a backup calibration lamp are collected at roughly monthly intervals and targeted

observations of well-calibrated surface vicarious calibration sites, such as Railroad Valley, Nevada are collected periodically when the site is cloud free.

The radiometric calibration of each spectral sample can be described in terms of a dark offset and a gain, both of which were extensively calibrated prior to launch (Rosenberg et al., 2016). The dark offset is defined as the response of a sample when no external illumination is entering the instrument. This component of response must be updated frequently in orbit because the

dark offset of each pixel is sensitive to small (millikelvin, mK) changes in the temperature of the FPA. The dark offset in the WCO2 and SCO2 channels is also sensitive to the temperature and associated black-body emission of the OBA.

To characterize dark offset changes, either summed or single-pixel measurements are collected for 1 to 3 minutes with the aperture door closed on every orbit where routine calibration measurements are collected. These dark data are used to update





the dark offset correction coefficients described by Rosenberg et al. (2016). For the forward, V7 L1B product, FPA and OBA temperature trends from the past 10 days (typically 146 orbits) are extrapolated to predict the dark offset corrections for the following week. For the retrospective product, V7r, temperature-dependent dark offsets corrections are derived using measurements acquired during the time period when the data were acquired.

Typically, OBA temperatures vary by around 100 mK each day due to orbit-to-orbit changes in spacecraft operations (Figure 4). FPA temperatures vary much more slowly ($< 5$ mK/day), but their temperatures increase as ice accumulates on the thermal straps that connect the FPAs to the cryocooler cold head (CH). Periodic adjustments as large as 1 K are made to the cryocooler CH temperatures to compensate for these changes. In the forward (v7) product, these large corrections require the dark offset calibration coefficients to be extrapolated far outside the training range. For most spectral samples, this extrapolation has

minimal impact because the dark offset is relatively insensitive to temperature, while a few samples have much greater temperature sensitivity (Figure 6). If their dark offsets are not accurately described by the extrapolated correction coefficients, errors can be introduced in the L1B product that can degrade the spectral fits performed by the L2 algorithm and introduce biases in the retrieved products. CH adjustments and other unanticipated FPA and OBA temperature corrections do not affect the retrospective, v7r products because they use interpolated, rather than extrapolated data to derive dark offset correction

coefficients.

In addition to dark offset variations, routine on-orbit calibration measurements have revealed changes in the radiometric performance due to increased numbers of bad pixels in the WCO2 and SCO2 FPAs, ice accumulation on the ABO2 FPA, and much smaller changes thought to be due to optical coating degradation. These phenomena and the methods used to characterize and mitigate their impacts are described in § 6.1 and § 6.5, respectively. To date, besides the effects of bad pixels, on-orbit

measurements provide no compelling evidence for changes in the overall sensitivity or gain linearity of the FPAs.

## 5.2   Spectroscopic Performance

The spectral dispersion and ILS of the OCO-2 instrument were well characterized and calibrated prior to launch (Frankenberg et al. 2014; Lee et al. 2016). The dispersion relation for each spectrometer specifies the mapping of the wavelengths of light onto specific pixels on the FPAs. The dispersion relation for each footprint in each spectral channel was derived prior to launch

(Lee et al., 2016) and is specified in terms of a $5^{th}$ order polynomial, the coefficients of which are listed in the L1B data files. Small changes in the dispersion coefficients were expected after launch due to Doppler shifts of the observed radiation (i.e., relative motion between the spacecraft and reflecting target). Dispersion changes can also be introduced by small displacements of the (cryogenically-isolated) FPAs relative the OBA, or small changes in the tilt of the gratings introduced by thermal gradients across the OBA. Changes in the focus of spectrometer collimator or camera assemblies could also produce

changes in both the dispersion and the ILS.

The spectroscopic performance has been routinely monitored in orbit using measurements of the positions and shapes of solar lines observed through the solar diffuser. Additional insight has been gained by monitoring changes in the fits to atmospheric spectra generated by the L2 retrieval process. These tests reveal small variations in the dispersion associated with changes in




the instrument thermal environment and with ice accumulation on the thermal straps that connect the cryocooler cold head to the FPAs. To date, these on-orbit measurements provide no compelling evidence of changes in the widths or shapes of the ILS functions derived prior to launch.

Data collected during the first 18 months of the OCO-2 mission indicate small dispersion changes that can be compensated by
changes in a wavelength offset or "shift" and wavelength "stretch" coefficients in the dispersion polynomial. The time evolution of the shift and stretch coefficients is shown in Figure 7. Prior to 2 July 2015, when nadir and glint observations were collected continuously for full, 16-day ground track repeat cycles, the instrument thermal environment equilibrated to distinct "nadir" and "glint" states that are clearly seen in the dispersion shift and stretch coefficients. Between June and November, when the observing mode switched between glint and nadir observations on alternating orbits, the instrument
thermal environment remained more stable over time. On 12 November of 2015, when the observing strategy was further optimized to always collect glint observations over ocean-only orbits, a new, stable thermal equilibrium was established.

In addition to its dependence on observing mode, Figure 7 shows that the dispersion also changes with time between instrument decon cycles. These small changes were not characterized prior to launch, but are thought to be driven by the following mechanism. As ice accumulates on the thermal straps, thermal losses between the CH and FPAs increase and the cryocooler
duty cycle increases to compensate. The accompanying increase in thermal load creates a thermal gradient across the cryocooler and the OBA structure, which increases with ice accumulation. This gradient introduces stresses on the thermal straps that connect the CH to the FPAs. These stresses introduce small ($< 1/10$ of the width of a pixel) lateral displacements (shifts) in the ABO2, WCO2, and SCO2 channels, with amplitudes that appear to depend on the distance between the FPA and the cryocooler CH. They also appear to produce small tilts in the FPAs, relative to the optical axis of the system, introducing
small changes in the dispersion stretch.

In the V7 product, the small changes in the dispersion shift and stretch coefficients are derived and corrected as part the spectral fitting process in the L2 retrieval algorithm. These changes are therefore not reported in the L1B V7 or V7r products delivered to the GES-DISC. Using data collected over the past 18 months, the OCO-2 calibration team has identified robust relationships between dispersion shift and stretch coefficients and CH and OBA temperatures that are routinely monitored. These insights
are being used to track dispersion changes so that they can be included in future versions of the L1B product. This should simplify the use of the L1B product by future users. The forward L1B product will include predicted dispersion shift and stretch values that are based on the instrument thermal environment. The retrospective L1B product will include refined estimates of the dispersion coefficients retrieved by the L2 algorithm.

## 5.3 Geometric Performance

To retrieve $X_{CO2}$ estimates with accuracies near 0.25%, the optical path length must be known to a comparable accuracy. An accurate geolocation of each sounding footprint is essential for defining the optical path length associated with each spectrum. To establish the geolocation of each sounding footprint, the location of the spacecraft along the orbit track, the pointing of the instrument boresight relative to a local coordinate system, and the relative pointing of the fields of view of the 8 footprints in



the 3 spectrometer channels must be known. The location of the observatory along its orbit track is determined by a Global Positioning System (GPS) sensor. The spacecraft orientation relative to fixed stars is determined by a star tracker. The spatial fields of view of each footprint and the relative bore sight alignment of the 3 spectrometer slits were characterized during the pre-launch instrument tests. In addition, a series of on-orbit calibration activities were implemented to verify the relative

alignment of the spectrometer bore sights, refine our knowledge of their alignment with the star tracker, and to provide an end-to-end verification of the geolocation software.

The relative alignment of the star tracker and the instrument boresight and the co-alignment of the instrument slits were tested with observations of the Moon. In these tests, the nominal center of the instrument field of view was pointed at an inertial point in space (a distant star field) that was expected to pass behind the gibbous lunar disk as the satellite traversed the night side of

the orbit.  These lunar tests verified that the instrument slits had not moved relative to each other by a measurable amount between the pre-launch testing and orbit.  They also indicated that the alignment of the star tracker and instrument was within specification and provided correction factors that were incorporated into the geometric calibration algorithm. The second step in the validation of the on-orbit geometric calibration involved observations of shoreline crossings for routine nadir and glint observations. This effort yielded geolocation errors no larger than half the size of the OCO-2 footprint.

## 15  6 On-Orbit Instrument Calibration Challenges

To fully exploit the information content of the high SNR OCO-2 spectra, they must be calibrated to a high degree of accuracy. For example, the relative radiometric performance (zero level offset, gain, and gain linearity) of spectral samples within a given channel must be known to within 0.1% across the spectral range of each channel to fully exploit the spectrally-dependent information in each sounding. The spectral calibration requirements (dispersion, ILS width, ILS shape) are equally demanding,

since uncertainties in these parameters lead to biases in retrievals (Connor et al., 2008, 2016). While the OCO-2 instrument was extensively characterized and calibrated prior to launch (Frankenberg et al., 2014; Rosenberg et al., 2016; Lee et al., 2016), a number of calibration challenges were encountered during the first few months of operation. These included increased numbers of bad pixels on the $CO_2$ FPAs, transient artifacts introduced by cosmic ray hits, spectral discontinuities for spatially inhomogeneous footprints, and time dependent changes in the sensitivity of the ABO2 channel. The methods used to identify,

characterize, and mitigate the impacts of each of these issues are described in the following sections.

### 6.1    Bad Pixels and Bad Samples

The FPAs used by OCO-2 are flight-spare units from the original OCO mission, which were delivered in 2006. A few percent of the 162,560 pixels in the 1016 by 160-pixel active area that is used to record science data are either dead, or respond to light or temperature in a way that is not consistent with the majority of the other pixels. These pixels must be identified and excluded

from the 20-pixel sums that are performed onboard or they will contaminate the 20-pixel spectral samples that constitute the spectra returned for each of the 8 footprints and three spectrometer channels. Spectral samples with too many bad pixels, or




with other known issues (e.g., optical artifacts on the short-wavelength ends of all 3 bands) are marked as "bad samples" in the L1B product (see variable L1BSc/InstrumentHeader/snr_coef), and should only be used with caution (or ignored) in higher-level processing.

The bad pixel identification process was started during prelaunch testing, but has to be updated periodically as the FPAs age.

Many additional bad pixels formed during the 2.5-year instrument storage period, whichwas imposed by the need to replace the OCO-2 launch vehicle after the loss of the Glory mission. The performance of other pixels has degraded in orbit due to cosmic radiation, thermal cycling of the FPAs, and other factors. The HgCdTe FPAs used in the WCO2 and SCO2 channels accumulated a large number of new bad pixels, particularly on the long wavelength ends of their spectral ranges while the instrument was in storage prior to launch (Figure 8). More than 2.5% of the pixels in the 160 by 1016-pixel area used for

recording science data are now classified as bad on these two FPAs (Table 2). In contrast, the ABO2 FPA developed very few new bad pixels while in storage. The rate of bad pixel formation in all 3 FPAs has been much slower since the observatory was launched and the FPAs were cooled to their operating temperatures. This experience is consistent with the experience of the James Webb Space Telescope (JWST) Detector Degradation Failure Review Board (Rauscher et al., 2012), which documented rapid degradation of this class of HgCdTe FPAs at room temperature, and much slower degradation at cryogenic

temperatures.

To identify bad or degraded pixels or spectral samples, the calibration team routinely collects dark frames and lamp "flat field" frames using "single pixel" mode, which returns all 225,280 pixels in the 220 by 1024 active region of each FPA. These single-pixel calibration data are analyzed to identify bad pixels, which are added to a "bad pixel map" for each FPA. This map is a 1024 by 220-byte array, where a value of 0 indicates that a pixel is good and a value of 1 indicates that a pixel's signal should

not be included in a spectral sample. Revised bad pixel maps are periodically uploaded to the spacecraft for use in the pixel-summing process.

As noted above, the spectral samples are created onboard by summing 19 to 21 adjacent pixels in a given FPA column. The algorithm is quite simple and does not apply bias or gain corrections to individual pixels prior to incorporating them into a sample. It does, however, assign different weights to pixels, based on their proximity to bad pixels. A single or double bad

pixel is replaced by the average of the closest adjacent normal pixels. If three or more adjacent pixels are bad within a single spectral sample, those values are excluded from the sum and the weights of the other pixels are renormalized. This is a good approximation because the spatial imaging of the telescope was intentionally blurred to yield a point spread function with a FWHM of 3 to 6 pixels. In this algorithm, a pixel within a given sample can therefore have a weight of 0 (bad), 1 (normal, with no adjacent bad pixels), 1.5 (one adjacent bad pixel), 2 (2 adjacent bad pixels), 2.5, or 3.

Early in the OCO-2 mission, the identification of bad pixels was a high priority because large numbers of bad pixels appeared on the $CO_2$ FPAs during the pre-launch storage period. These bad pixels produced artifacts in the spectra that compromised their utility. As the bad pixel maps improved, a larger fraction of the spectral samples in each spectrum could be used to generate higher level products. Because the ABO2 FPA did not degrade significantly during the pre-launch storage period, its bad pixel map has not been updated since launch. In contrast, the bad pixel maps for the two $CO_2$ FPAs were updated four





times between September 2014 and March 2015 as the bad pixel population was better characterized (Table 2). The bad pixel map version used for each granule is specified in the metadata of each file in the variable "BadPixelMapVersionNum" [1].

In addition to the bad pixel map updates, the calibration team also maintains a bad sample list, which is applied later in data processing to mask additional spectral samples with inadequately corrected bad pixels. This approach provides more flexibility

than the bad-pixel maps alone, because it does not permanently mask pixels whose performance is better calibrated through further analysis. The bad sample lists are updated with each bad pixel map update, or when a sample's performance is altered by a cosmic ray event, decon thermal cycle, or other factors. The bad sample list is also used to mask spectral samples at the edges of the FPA where the ILS could not be adequately characterized, and samples at the short wavelength end of each band that are compromised by optical distortion. These latter two factors account for the majority of the samples included in the bad

sample list.

**Table 2. Bad Pixel Map Version and Number of bad pixels.**

| Start Date | Bad Pixel Map Version ABO2/WCO2/SCO2 | Start Orbit | ABO2 # Bad Pixels | WCO2 # Bad Pixels | SCO2 # Bad Pixels |
|---|---|---|---|---|---|
| 3/30/2012 | 5/5/5 | pre-flight | 853 | 1606 | 1400 |
| 9/5/2014 | 5/6/6 | 955 | 853 | 2725 | 2164 |
| 11/10/2014 | 5/8/8 | 1913 | 853 | 3631 | 2859 |
| 11/21/2014 | 5/9/9 | 2074 | 853 | 4371 | 3699 |
| 2/13/2015 | 5/10/10 | 3297 | 853 | 4520 | 4414 |

The most significant change to the bad sample list was made on 10 November 2014 (Table 3), when 565 and 517 samples

were removed from the list in the WCO2 and SCO2 channels, respectively, based on further characterization of in-flight data. The only change to the ABO2 channel was the addition of 5 samples to the list on 13 February 2015. In addition, the SCO2 bad sample list was increased by a single sample each time on 7 July 2015, 13 October 2015, and 10 November 2015 to respond to new bad samples caused by cosmic ray events. The WCO2 and SCO2 channels were changed significantly (adding 49 and 155 samples to the bad sample list, respectively) on 1 January 2016 in response to improved sample dark correction.

On 1 March 2016, the bad sample list was updated to add one more sample to the list in the SCO2 channel. A complete list of bad sample changes is provided in Table 3, where we list the number of samples rejected, out of the 8128 available in each channel. The distribution of bad samples is shown as a function of wavelength (L1B column), footprint, and channel in Figure 9.

**Table 3. Bad Sample List Change History**

---

[1] Note: the bad pixel map is applied only to summed spectral samples. It is not applied to the full-resolution color slices.



| Start Orbit | ABO2 Samples | WCO2 Samples | SCO2 Samples | Reason | Date |
|---|---|---|---|---|---|
| 958 | 2498 | 2241 | 2471 | BPM 6 | 2014-09-06 |
| 1913 | 2498 | 1676 | 1954 | BPM 8 | 2014-11-10 |
| 2074 | 2498 | 1695 | 1934 | BPM 9 | 2014-11-21 |
| 3297 | 2503 | 1676 | 2196 | BPM 10 | 2015-02-13 |
| 5297 | 2503 | 1677 | 2196 | Cosmic Ray Event | 2015-07-01 |
| 6820 | 2503 | 1677 | 2197 | Cosmic Ray Event | 2015-10-13 |
| 7227 | 2503 | 1677 | 2198 | Cosmic Ray Event | 2015-11-10 |
| 7977 | 2503 | 1726 | 2353 | Sample Dark Correction | 2016-01-01 |
| 8851 | 2503 | 1726 | 2354 | Single New Bad Sample | 2016-03-01 |

With each new bad pixel map, new bad sample lists and calibration tables were constructed. Each new bad pixel required modifications to the radiometric gain of its spectral sample. For V7 L1B products, the gain and offset adjustments were extensively tested by performing L2 retrievals on a large test data set to identify persistent outliers. Spectral samples that continued to produce large spectral residuals were added to the bad sample list. This method for identifying new bad samples has been refined over the first 18 months of the mission and is now a routine part of the L1B validation process (Table 3).

## 6.2    Transient Cosmic Ray Artifacts

Cosmic rays rarely produce permanent damage to the OCO-2 FPAs, but energetic particles with a range of energies produce ion trails as they traverse the FPAs. Because the electrons in these ion trails are indistinguishable from those produced by photons, they produce spurious intensity spikes in individual exposures. The largest cosmic ray spikes are seen in the ABO2 channel, because the silicon photodiode array used in the ABO2 FPA is physically thicker (~100 μm; Bai et al., 2008) and more sensitive to cosmic rays than the thinner (5-10 μm; Beletic et al., 2008) cadmium-zinc-tellurium (CdZnTe) substrates used for the FPAs in the $CO_2$ channels.

Primary and secondary cosmic rays with a broad range of energies are occasionally seen everywhere along the orbit path, but they are most common in the vicinity of the South Atlantic Anomaly (SAA), where the base of the inner Van Allen radiation belt extends to altitudes below the 705 km orbit altitude of OCO-2 (Stassinopoulos and Raymond, 1988). The approximate geographical extent of the SAA impacts on the OCO-2 sensors can be seen in Figure 10. Over this region, up to 2% of the 1024 spectral samples in an $O_2$ A-band spectrum can be contaminated in a single 0.333 second exposure. An example of cosmic ray artifacts in a spectrum is shown in Figure 11, where large cosmic ray spikes are seen above the A-band continuum across the entire spectrum. All but the brightest of these artifacts are eliminated when the FPA is reset for the next exposure. An algorithm was developed to identify and screen cosmic ray artifacts in the L1B products delivered to the science community (Eldering et al. 2015). For each orbit granule, this algorithm performs a singular value decomposition of all spectra collected





in each footprint in each band. A least squares approach is then used to fit the 40 leading eigenvectors to each spectrum and then identify the spectral samples that are outliers from the fit. These outliers (positive spectral residuals) are flagged, based on the number of standard deviations between the fit and the recorded spectrum for each sample. The degree of disagreement is recorded in the L1B file, in the group "SpikeEOF".

The OCO-2 L2 retrieval algorithm uses the weighted residuals in the L1B file to flag individual spectral samples as "contaminated." For the V7 product, samples are assumed to be contaminated if their weighted spectral residual exceeds +6 σ, and if the sounding falls within a geographic region around the SAA (−50° to 0° latitude, −90° to 10° longitude). A one-sided threshold value is used because cosmic ray hits can only cause a positive anomaly in measured spectra. The V7 L1B ATBD (Eldering et al. 2015) advises users to adopt a similar approach for flagging spectral samples contaminated by cosmic

rays and to apply this threshold only in the vicinity of the SAA. The 7 cloud screening (Taylor et al., 2016) did not incorporate a cosmic ray flag, and thus often failed to converge for many soundings in the SAA region. These soundings were flagged as bad and not processed in this version of the product. To address this issue, a cosmic ray screening step will be performed before running the cloud screening algorithms in future data products.

## 6.3    Radiance Discontinuities due to FPA Rotation

For imaging grating spectrometers, the spectrometer slits, the grooves on the diffraction gratings, and columns of the FPAs must be well aligned, and other optical distortions (keystone, smile) must be minimized to ensure that a fixed series of rows on the FPA will sample the same angular field of view (or spatial footprint) throughout the spectral range recorded by the FPA. For the OCO-2 instrument, perfect alignment (clocking) of the FPAs with the other optical components was precluded by a physical obstruction discovered late in the instrument assembly process. The focal plane arrays are therefore slightly rotated,

or "clocked," with respect to the slit and grating. Consequently, a given geographic position across the slit does not map onto a single row of pixels on the FPA, but instead varies, roughly linearly, with spectral position across the FPA.

To compensate for the rotation of the OCO-2 FPAs, and record the same spatial information across the entire spectral range, the starting row index for each of the 1016 spectral samples corresponding to a given footprint can be adjusted in increments of one pixel. The columns where these clocking adjustments occur are recorded in the L1B data files delivered to the

community (variable name: clocking_shift_color_indicator) and listed in Table 4. A single pixel corresponds to about 1/20th of the area of a summed footprint (Figure 2). Adjusting the footprint boundaries in single-pixel increments works well in spatially homogenous scenes, but can produce radiance discontinuities in spectra of scenes with strong intensity variations, especially if large spatial gradients in brightness occur near the edge of a footprint. In such scenes, the radiance discontinuities will typically have opposite signs in adjacent footprints. These clocking discontinuities are corrected as part of the L1B

calibration process.

**Table 4. Clocking adjustment columns in each spectral channel**





| Channel | ABO2 | WCO2 | SCO2 |
|---|---|---|---|
| Clocking Adjustment Columns | 123, 363, 603, 843 | 51, 475, 899 | 87, 211, 335, 459, 583, 707, 831, 955 |

The clocking correction algorithm enforces continuity in radiance across single pixel footprint shifts to compensate for spectral discontinuities associated with single-pixel adjustments (Figure 13). This algorithm exploits the spatial information provided by the single pixel color slices described in §2. Although the spectra recorded for each spatial footprint consist of 1016, ~20-pixel "spectral samples," the color slices record the spatial information at single pixel resolution across each footprint for up to twenty of the 1016 columns (wavelengths) across each FPA. The clocking correction algorithm uses the single-pixel radiances recorded in the color slices to map spatial variation across the spectral range, particularly near those columns where clocking jumps occur (Figure 14a).

As noted in §2, the locations (columns) for the color slices on each FPA can be changed by commands from the ground. The most recent update was 17 October 2014 (orbit 1567). The color slice positions are listed in the L1B data files (see variables: ColorSlicePositionO2, ColorSlicePositionStrongCO2, ColorSlicePositionWeakCO2). In principle, placing the color slices adjacent to the columns with the clocking jumps would be ideal, but this is sometimes problematic because some of these columns include sharp atmospheric or solar absorption features, and these features can be shifted by up to 3 columns by Doppler shifts associated with the spacecraft motion and observing geometry. If a color slice is placed in a spectral region where the radiance changes rapidly (e.g., near an absorption line), spectral variations could be misinterpreted as artifacts associated with spatial inhomogeneity across a footprint (Figure 14). Therefore, the color slices used by the clocking correction algorithm were located in columns where the spectral variations were small (i.e., in continuum regions) wherever possible.

The availability of continuum regions varies substantially from channel to channel. For the ABO2 FPA, continuum regions were selected just beyond the weaker absorption lines near the edges of the FPA as well as in the continuum near the center of the band. A similar approach was adopted for the WCO2 channel. The deeply saturated SCO2 channel offers no true continuum regions, so the color slices used by the clocking algorithm are located in the most slowly varying regions available near the edges and center of the $CO_2$ band.

The number of clocking adjustments needed in each channel depends on the rotation angle of the FPA relative to the slit. It therefore varies from channel to channel, ranging from 8 discontinuities in the SCO2 channel to only 2 in the WCO2 channel (Figure 12). Figure 14 illustrates the choice of color slices for the ABO2 channel. In these figures, only those color slices that are labeled by a column index are used by the clocking algorithm. Other color slices have been located within strong absorption features to assess the spatial inhomogeneity due to clouds and aerosols or to monitor solar lines used in the Solar Induced Chlorophyll Fluorescence (SIF) retrieval.

Figure 15 shows an example of a spatial illumination variation within a single ABO2 frame. Footprints 4 and 5 show significant differences in both radiance and slope between their left and right boundaries. These large radiance gradients can produce large clocking errors if not corrected. To correct these errors, the clocking correction algorithm employs the following steps:



1) Single pixel radiance measurements from color slices are used to characterize spatial gradients across each footprint at several wavelengths across the spectral range sampled by the FPA. This is done by calculating ratios of the footprint-averaged radiances computed by shifting the footprint boundaries in one-pixel (row) increments (after correcting bad pixels in each color slice).

2) Radiance ratios from groups of color slices in nearby columns are combined to increase the signal to noise ratio of each radiance ratio.

3) The radiance ratios for each color slice group are then interpolated across the entire spectral range using a piecewise linear function.

4) This piecewise-linear function is then used to define a correction function, $C(f, \lambda)$, that is multiplied by the radiance returned in each spectral sample (e.g. for each footprint, $f$, and each column, $\lambda$).

Because the clocking introduces radiance variations that propagate almost linearly across the wavelength domain of each FPA, the correction function has a saw-tooth shape, with discontinuities at each clocking step. This function is normalized to unity across the spectral domain, so that it does not introduce a shift in the mean radiometric calibration for the spectral channel. A detailed, step-by-step description of the clocking correction algorithm is given the OCO-2 Level 1B Algorithm Theoretical Basis Document (Eldering et al. 2015), along with an analysis of its performance for a typical orbit.

Clocking errors are most easily seen in differences between the observed spectra and best-fit synthetic spectra produced as part of the L2 data product generation. Figure 16 shows spectral residuals in all three spectral channels before (black) and after (red) the clocking correction is applied to a footprint with a relatively large radiance variation. The clocking errors appear as sharp discontinuities prior to the correction, but are virtually invisible after the correction. In Figure 16a, the maximum amplitudes of the corrections are ~1.8, 2.9, and 2.1% in the ABO2, WCO2, and SCO2 channels, respectively. The goodness of fit, expressed in terms of the ratio of the root-mean square residuals to the random error, $\chi^2$, was also improved after the correction, but their relatively large post-correction values indicate that the spatial radiance variations may have been introduced by a sub-footprint scale cloud. For cloud-free scenes over land, clocking corrections are typically smaller than 1%. Figure 16b shows a case where the clocking correction worked well for the ABO2 and SCO2 channels, but under-corrected the SCO2 channel. These scenes are usually screened by the OCO-2 L2 retrieval algorithm.

## 6.4    Polarization Orientation Anomaly

While sunlight incident on at the top of Earth's atmosphere is not polarized, scattering by the atmosphere and surface can introduce varying amounts of polarization in the reflected sunlight observed by OCO-2. The degree of polarization can vary substantially, but is typically greatest at the Brewster angle over the ocean (~53° solar incidence angle) or near the terminators, where molecular (Rayleigh) scattering by the atmosphere introduces significant polarization. While the degree of polarization is typically unknown, the angle of polarization can be derived from the illumination and observing geometries. For example, if we define a "principal plane" that includes the sun, the surface footprint, and the instrument aperture, the light polarized



parallel to this plane (e.g. the P polarization) is much more strongly attenuated than the light polarized perpendicular to this plane (e.g. the S polarization; Crisp et al., 2008).

Diffraction gratings for high resolution grating spectrometers, like those used by OCO-2, can also introduce significant polarization in the light that they disperse. If the reflected sunlight is strongly polarized and the angle of polarization of the

incoming light is orthogonal to the angle of polarization diffracted most efficiently by the gratings, the signal can be substantially reduced. This issue was recognized early in the design of the original Orbiting Carbon Observatory (OCO) instrument.

Unfortunately, the polarization angle most efficiently diffracted by the gratings was misinterpreted by the instrument designers. They reported that the instrument was most sensitive to light polarized parallel to the long axis of the slits (Pollock et al.,

2010). If this were the case, the largest signals would be obtained if the slits were aligned perpendicular to the principal plane (i.e. parallel to the surface).  The spacecraft design and mission observing strategy were optimized to exploit this result (Crisp et al., 2008).  In fact, like most grating spectrometers, the OCO and OCO-2 spectrometers are most sensitive to light polarized perpendicular to the long axis of the slits (e.g. in the direction of dispersion).

This design flaw was not caught in the pre-launch instrument assembly and testing program for OCO because the instrument

team was focusing on maximizing the throughput of the instrument, and was not considering the satellite viewing geometry or polarization characteristics of the surface and atmosphere. This error was not caught for OCO-2 because the instrument and spacecraft were "build to print" copies of the original OCO, and similar pre-launch integration and test procedures were followed (Rosenberg et al. 2016).

This design error was discovered by the science team almost immediately after launch. It was first seen in the earliest target

observations of Total Carbon Column Observing Network (TCCON) sites, where it introduced large viewing-angle-dependent biases in the retrieved properties. These biases vanished if Rayleigh scattering was ignored or if the assumed polarization angle was artificially rotated by 90°. Soon afterwards, the polarization orientation error was seen in the earliest glint observations over the ocean. These observations showed that the radiance levels were dramatically lower than predictions obtained from the ocean surface reflectance model (Cox and Munk, 1956) at solar zenith angles between 50° and 60°, near the Brewster angle

for sea water. Individual spectral samples were expected to yield continuum SNR values between 400 and 1000 at these angles, but values much less than 100 were seen (Figure 17). Note that SNR values near 200 are needed to yield $X_{CO2}$ estimates with single sounding random errors less than 1 ppm.

Once the root cause was identified, a range of mitigation options were studied, and one of the simplest was identified and implemented. Because the OCO-2 spectrometers are the only payload on the spacecraft, the observing geometry was modified

to rotate the entire spacecraft by 30° (clockwise from above) about the center of the instrument boresight for glint observations. This corresponds to the "yaw angle" of the spacecraft. With this 30° yaw angle, the SNR values over the ocean were restored to values between 200 and 600 at latitudes spanning the Brewster angle (Figure 18). This yaw angle reduces the amount of sunlight incident on the solar panels, but the power system had ample margin to support this change. The Level 1B products





now reflect this modified yaw angle, polarization angle, and footprint-dependent Stokes coefficients. For nadir observations, the spacecraft initially continued to operate with the slit oriented perpendicular to the principal plane because these observations returned adequate signal only over land, where the reflected sunlight is not strongly polarized. On 12 November 2015, a 30° yaw was also adopted for nadir observations to minimize the thermal perturbations introduced to the OBA in

switching between nadir and glint observations.

## 6.5    ABO2 Sensitivity Variations

As the radiometric calibration record extended through the first few months of the mission, a trend in the ABO2 channel sensitivity was detected in the routine solar and lamp calibration observations. Little or no change was seen in the $CO_2$ channels (Figure 18). Comparisons of solar diffuser observations with the lamp and lunar calibration observations appeared to confirm

the basic features of these sensitivity variations. The ABO2 sensitivity appeared to decrease at an accelerating rate while the instrument's OBA and FPAs were at their operating temperatures. The sensitivity of the ABO2 channel was largely restored after each decon cycle.

An investigation was initiated to identify the root cause of the observed ABO2 signal variations. Both optical contamination and electrical sources were investigated. Water ice accumulation on the FPAs was one of the first optical contaminants

considered, but this hypothesis was initially discounted because water ice absorbs much more strongly in the spectral ranges of the $CO_2$ channels, which showed much less degradation. A very thick layer of ice (many tens of microns) would be needed to produce the observed absorption on the ABO2 FPA. There was no plausible mechanism that would deposit much more water ice (or any other known optical contaminant) in the ABO2 channel than in the $CO_2$ channels. In addition, a thick layer of ice would significantly alter the ABO2 instrument line shape (ILS) function, and no changes in the ILS were apparent. A

variety of electrical mechanisms for reducing the sensitivity of the FPAs or their signal chains were also considered, but then discarded as tests and analyses progressed. Meanwhile the OCO-2 calibration team developed an empirical correction for the ABO2 sensitivity variations and implemented this correction in the V7 L1B products delivered to the GES-DISC and used to generate the V7 L2 products.

As additional data were collected, other features of the ABO2 degradation were identified. In particular, while the signal

degradation was largely restored to its pre-launch value by each decon cycle, the signal did not recover to the pre-launch value. Instead, the recovery level fell slowly over time. This suggested that there may be at least two, possibly independent sources for the ABO2 sensitivity variations. The first resulted in a rapid signal decay that accelerated with time, but was largely corrected when the instrument OBA and FPAs were warmed to near room temperature during decon activities. The second was slow and monotonic, and was not affected by the decon cycles. These "fast" and "slow" contributions to the sensitivity

degradation were studied in parallel.



### 6.5.1    The Fast Degradation

As the investigation proceeded, the discovery of what appeared to be ice crystals forming on the ABO2 FPA during pre-launch testing once again brought further scrutiny to this mechanism as a possible source of the "fast" degradation. A plausible mechanism was soon identified. The ice was not reducing the sensitivity of the ABO2 FPA by absorbing the incident light,

but by degrading the anti-reflection (AR) coating on the FPA. The ABO2 FPA is made of silicon, which has a relatively high refractive index (~3.7) and a high first-surface reflectance (~30%) at the wavelengths covered by this channel. To improve its sensitivity, the vendor had added a thin (~120 nm thick) AR coating to the front surface of the FPA, reducing its reflectance to < 10%. The AR coating has an index of refraction that is similar to that of water ice. As ice accumulated on the FPA, and its thickness approached that of the AR coating, the surface reflectance increased at an increasing rate. When the FPAs were

warmed to room temperature during a decon cycle, the ice sublimated, and the performance of the AR coating was restored. This mechanism explains the observed depth and rate of the "fast" component of the sensitivity degradation, and its recovery. The very thin ice coating on the ABO2 FPA (less than 1/180[th] the width of a pixel) helps to explain why the fast degradation was not accompanied by a significant change in the ABO2 ILS or other aspects of the instrument performance. More recently, a small zero level offset, with an amplitude as large as 0.8% of the continuum, has been seen when the ABO2 signal degradation

is at its maximum. This change was first detected in the core-to-continuum ratios of the solar Fraunhofer lines used to measure Solar Induced Chlorophyll Fluorescence (SIF, Figure 19). It has since been seen in the routine lamp and lunar calibration observations. This zero level offset probably occurs as some of the light reflected by the degraded AR coating on the ABO2 FPA is reflected back to the FPA by the spectrometer optics. It is currently being characterized and will be corrected in future versions of the OCO-2 L1B product.

This mechanism for the fast degradation provides some insight into why much smaller changes are seen in the $CO_2$ channels. The HgCdTe FPAs used in the WCO2 and SCO2 channels have a lower intrinsic surface reflectance and use thicker, higher index AR coatings than those used in the ABO2 channel. A much thicker layer of ice is therefore needed to degrade the performance of the AR coatings these channels.  A small amount of degradation has been detected in the sensitivity of the WCO2 channel (see the green line in Figure 18) that may also be caused by this mechanism.

### 6.5.2    The Slow Degradation

While a thin layer of ice accumulating on the ABO2 FPA provides a plausible mechanism for the "fast" degradation, it does not explain the slow, monotonic ABO2 signal loss seen in the solar diffuser and lamp measurements, which is not corrected by the instrument decon activities. The solar calibration observations indicate that the amplitude of this contribution exceeded 5% during the first 18 months of the mission, but the rate of change seems to be decreasing with time (Figure 20a). An in-

depth analysis of lunar calibration observations, however, indicates a much smaller rate of signal loss than that inferred from the solar and lamp data (Figure 20b). These measurements, which are made using the science aperture (without the diffuser), indicate that only about one-fifth of the observed attenuation can be attributed to reductions in the throughput of the telescope





and spectrometers. Most (~80%) of the apparent slow degradation appears to be caused by a reduction in the throughput of the solar diffuser, perhaps associated with degradation of the gold coatings on its active inner surfaces. The remaining ~1% decrease in signal over the first 18 months in orbit is most likely due to degradation of the gold coatings in the telescope mirrors and other optical components. This source of signal loss was anticipated prior to launch and was expected to have its largest

impact on the ABO2 channel. However, its actual rate was impossible to predict.

The gold coatings on the lamp diffuser (on the back of the aperture cover) are much more exposed to the space environment than those in the solar diffuser and spectrometer optics, and show larger reductions in reflectivity since launch (see square boxes in Figure 18). The output of the calibration lamps has also decreased somewhat in the ABO2 channel. Fortunately, the rate of decay of the reflectance of these coatings and the lamps appears to be decreasing with time and can be adequately

characterized with lunar observations and observations of well-characterized vicarious calibration sites.

While these processes are correctable, and do not threaten mission success, they do reduce the accuracy of the absolute radiometric calibration of the V7 L1B products. The initial ABO2 degradation correction that was applied to this product did not discriminate between the "fast" and "slow" degradation. This empirical correction assumed that both the fast and slow components of the signal loss were due to changes in the throughput of the instrument's science optical path or the sensitivity

of its FPAs. By including the slow degradation of the solar diffuser in this correction, the radiometric calibration process has over-corrected L1B products since launch. This error has recently been confirmed by lunar calibration data and in observations of parts of the Sahara and Arabian deserts, which appear to be brightening over time in the OCO-2 product, relative to nearly coincident results collected by the Moderate Resolution Imaging Spectroradiometer on the NASA Aqua platform (MODIS/Aqua). This error is currently being more fully characterized using lunar observations, vicarious calibration

observations of Railroad Valley, NV, and through additional comparisons with observations collected by MODIS/Aqua and the Japanese Greenhouse gases Observing SATellite (GOSAT, nicknamed "Ibuki"). It will be corrected in future versions of the OCO-2 L1B product.

## 7 Conclusions

OCO-2 was successfully launched on July 2, 2014. Two months later, it began routinely returning almost one million soundings

over the sunlit hemisphere each day. While its 3-channel, imaging grating spectrometer was extensively characterized and calibrated prior to launch (Frankenberg et al. 2014; Rosenberg et al. 2016; Lee et al. 2016), and is returning high quality data a number of subtle calibration challenges were identified during its first year and a half in orbit. To address these issues, the instrument calibration and observing strategy have been continuously refined during this period.

The OCO-2 team began delivering its first global data product, V7, to the GES-DISC in early June 2015 for distribution to the

science community. This product includes calibrated, geo-located spectral radiances (L1B products), and retrieved geophysical quantities, including spatially resolved estimates of $X_{CO2}$, surface pressure, and solar-induced chlorophyll fluorescence (L2 products). Updates for bad pixels, cosmic rays, clocking errors, and some aspects of the ABO2 sensitivity variations have been





incorporated into the V7 L1B product. The impacts of a polarization anomaly were mitigated through changes in the measurement approach. Other issues, such as thermally-induced changes in the instrument dispersion are not being corrected in the L1B product because a dispersion correction is a routine part of the L2 processing. These corrections will be incorporated into future versions of the L1B product. Corrections for other calibration issues, such as the zero level offset associated with

the ABO2 "fast" degradation and the slow degradation of the solar diffuser are under development, and will also be incorporated into future L1B products.

These updates to the instrument calibration have fully exploited the on-board calibration system, as well as observations of the Sun, the Moon, and well calibrated "vicarious calibration" targets on the Earth's surface, such as Railroad Valley, NV. Comparisons with nearly coincident observations collected by MODIS/Aqua and GOSAT are also being used to diagnose and

reduce calibration uncertainties. The impact of these uncertainties on the L2 products are being evaluated using comparisons with observations from the TCCON network and other standards. These comparisons indicate that OCO-2 returns $X_{CO2}$ estimates with single-sounding random errors between 0.5 and 1 ppm at solar zenith angles as large as 70 degrees. The median difference between OCO-2 and TCCON $X_{CO2}$ estimates is less than 0.5 ppm and the root-mean-square (RMS) differences are typically < 1.5 ppm (Wunch et al., 2016).

### Acknowledgements

Part of the research described in this paper was carried out at the Jet Propulsion Laboratory, California Institute of Technology, under a contract with the National Aeronautics and Space Administration. The CSU/CIRA contribution to this work was supported by JPL subcontract 1439002.

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

.

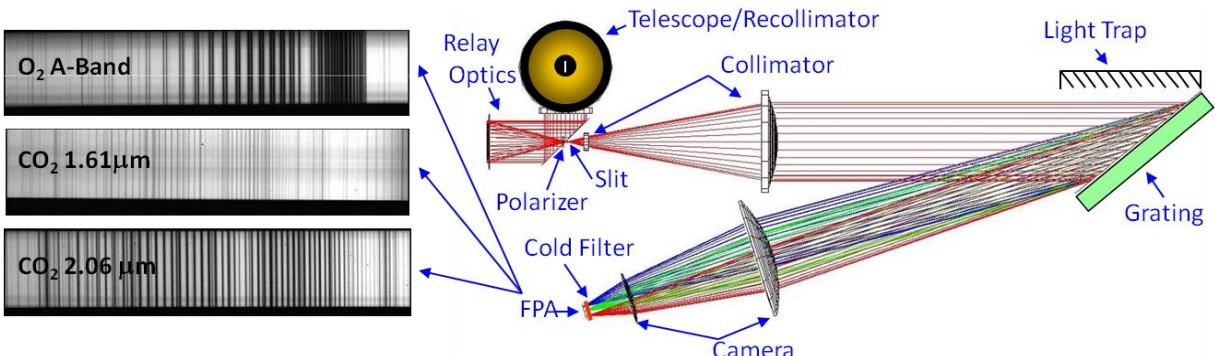

**Figure 1: The OCO-2 instrument incorporates three imaging grating spectrometers with similar optical layouts. The major optical components and optical path for a single channel are shown on the right hand side. The orientation of the narrow entrance slit is indicated by the thin, vertical white line in the secondary mirror at the center of the Telescope/Re-collimator aperture. Images of spectra recorded by the focal plane arrays (FPAs) in the 3 spectral channels are shown on the left. A spatially-resolved "image" is formed along the slit, while the spectra are dispersed perpendicular to the slit, with wavelength decreasing from left to right.**





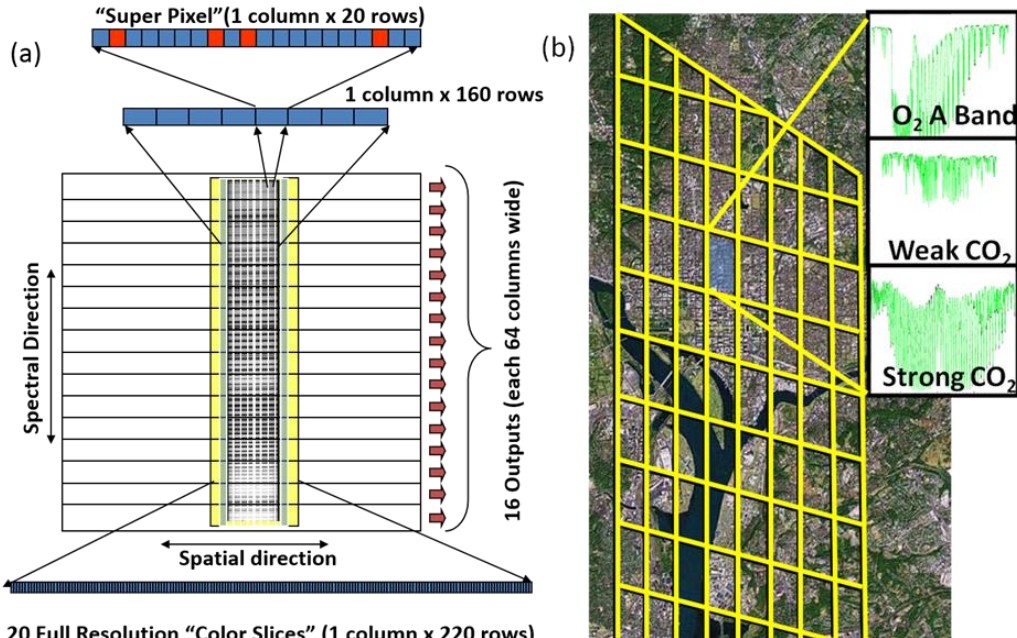

**Figure 2: The illumination and readout approach used for the OCO-2 FPAs, showing the direction of spectral dispersion from bottom to top, and the spatial direction from left to right. The ~160 illuminated pixels in the spatial dimension are summed into eight 20-pixel "footprints." If one or more of the pixels in a footprint is "bad" (red pixels), it is eliminated from the sum. If one or two contiguous pixels in a column are bad, then they are replaced with the average of their good neighbors. If three or more bad pixels are contiguous, then they are replaced with zeros. This algorithm is only applied in the spatial direction—spectral information is never mixed. One of the 20 full-resolution "color slices" is also shown at the bottom. (b) Spatial layout of 8 cross-track footprints for nadir observations over Washington, D.C. Each footprint is shaped like a parallelogram, rather than a square, because of the rolling readout of the FPAs and the spacecraft motion.**





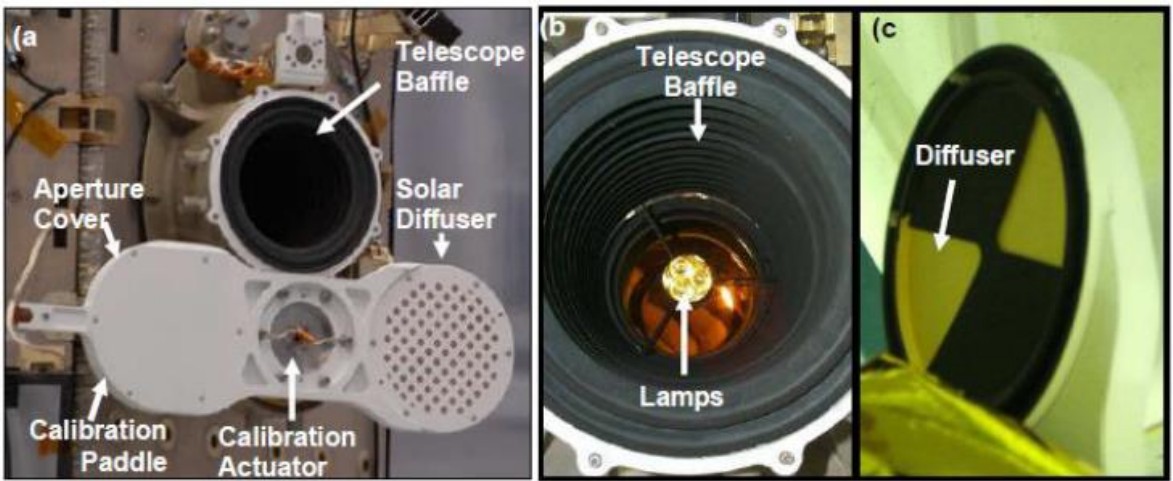

**Figure 3: The On-board Calibrator (OBC) is attached to the telescope baffle assembly. (a) The OBC is shown in the "open" position for taking science data, with the aperture cover is to the left and the solar diffuser to the right. (b) The telescope baffle assembly is shown along with the incandescent lamps used to illuminate the diffuser on the back of the aperture cover. (c) The inner surface of the aperture cover is partially coated with a roughened gold surface that is illuminated by the lamps to produce a spatially uniform illumination field.**





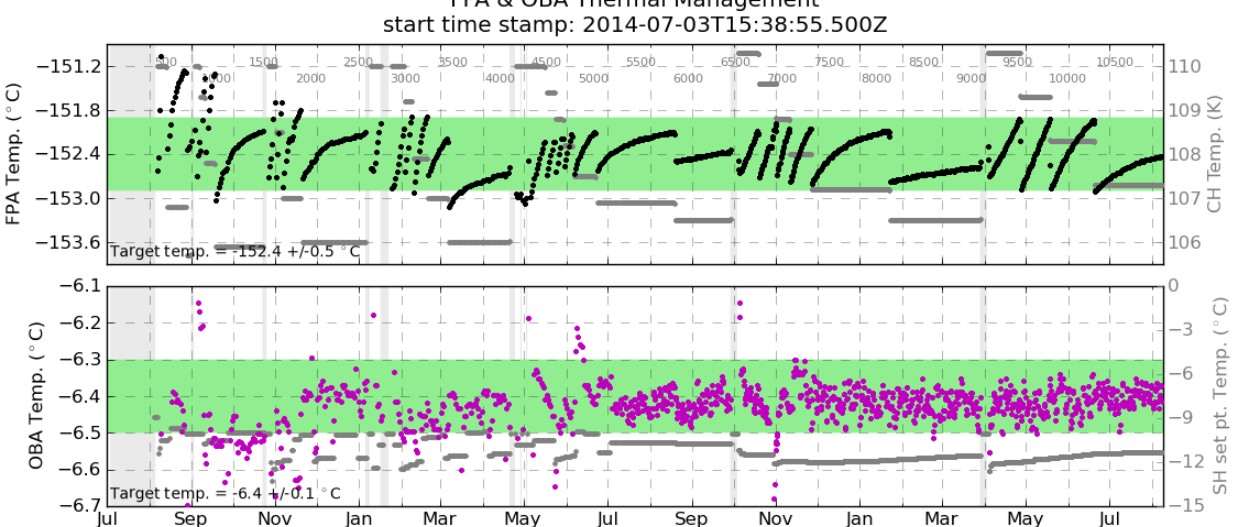

**Figure 4: Thermal performance of the OCO-2 instrument FPAs (top) and OBA (bottom) are shown as a function of time for normal operations. The FPA temperatures (black dots in top plot) are maintained by controlling the temperature of the cold head (CH) of the cryocooler. The CH temperatures are shown as grey points (that appear as horizontal grey lines) in the top plot, and their values are labeled on the right hand y-axis. During routine operations, the CH temperatures must be adjusted as ice accumulates on the thermal blankets that insulate the thermal straps linking the FPAs to the CH, reducing their efficiency. OBA temperatures (pink dots in bottom plot) are maintained by controlling the temperature of the thermal shroud that encloses the OBA. The set point of the shroud (SH set pt.) is shown as grey points on the bottom plot and their values are labeled on the right hand y-axis of that plot. The light grey vertical bars indicate decontamination cycles when normal operations are suspended and the FPAs and OBA are raised to near room temperature to remove ice from the thermal straps and FPAs.**





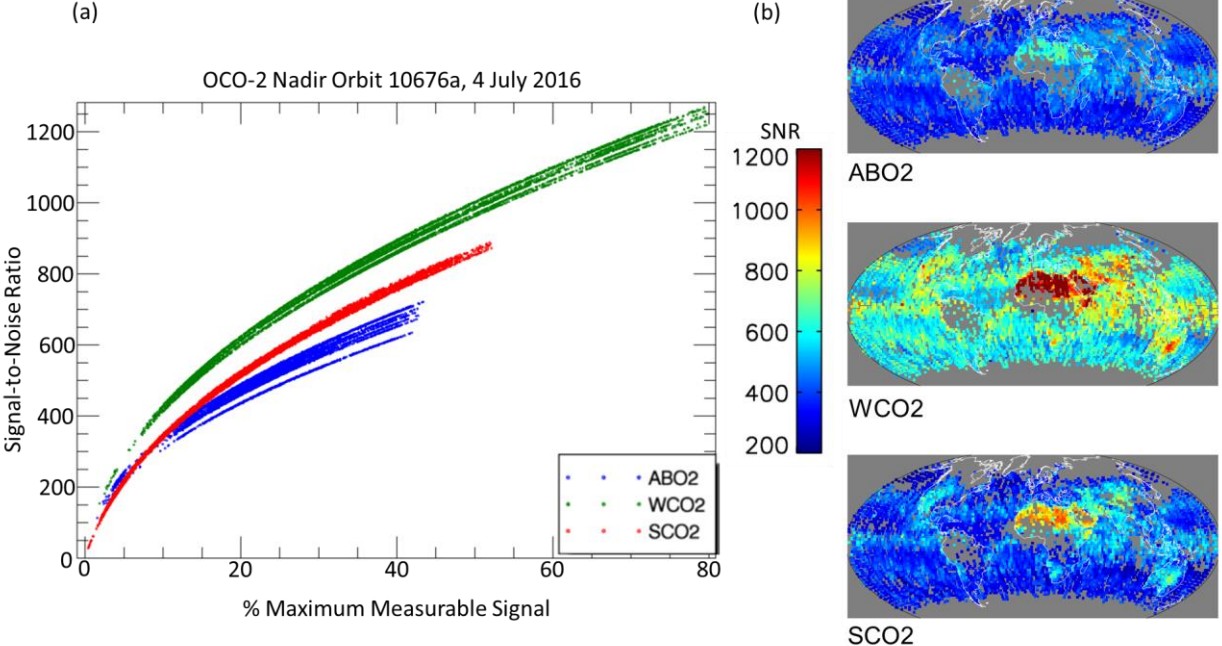

**Figure 5: (a) Median Signal-to-Noise Ratios (SNR) for individual soundings in the ABO2 (blue), WCO2 (green), and SCO2 (red) channels are shown as a function of the % maximum measurable signal for a nadir orbit observed on 4 July 2016. The parallel traces show results for the 8 footprints in each channel. (b) The mean continuum single-sounding signal-to-noise ratios (SNR) in 2° x 2° bins is shown for the ABO2 (top), WCO2 (middle), and SCO2 (bottom) channels for April 2015.**





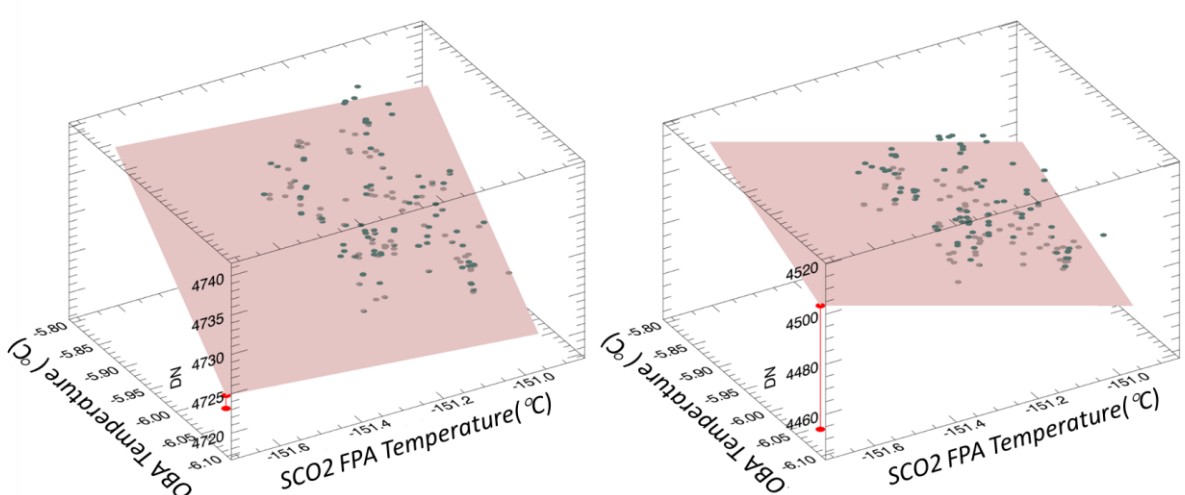

**Figure 6: The dark coefficient fitting process is illustrated for a typical "well-behaved" spectral sample (left) and a "temperature-sensitive" spectral sample (right) in the SCO2 channel. In both cases, individual spots show the sample response (expressed in digital numbers, DN) as a function of the temperature of the FPA and the OBA. The fitting process assumes that the dark response is a linear function of the FPA and OBA temperatures over the training range. The dark response of the well-behaved sample changes by 4.7 DN/°C with respect to the FPA and 89.9 DN/°C with respect to the OBA. The dark response of the temperature sensitive sample changes by 47.2 DN/°C with respect to the FPA and 126.3 DN/°C with respect to the OBA. If the FPA or OBA temperatures move outside of this training range, the dark response of the temperature-sensitive sample is much less predictable than that of the well-behaved sample.**





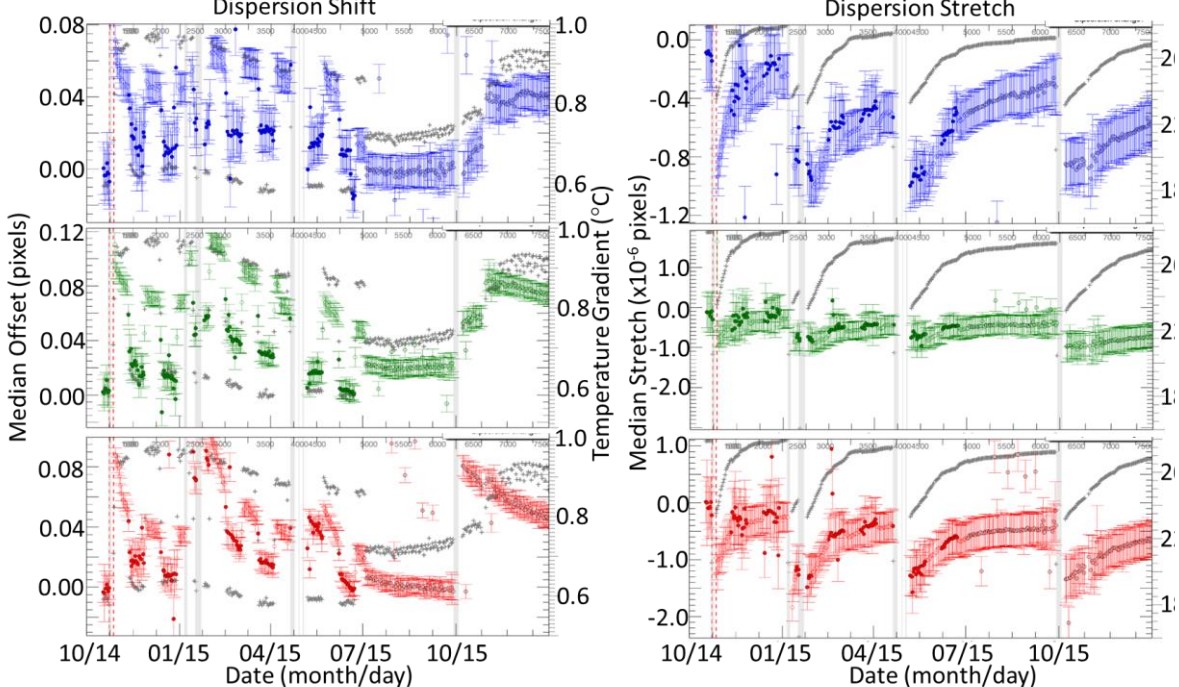

**Figure 7: The time evolution of the dispersion shift/offset (left column) and dispersion stretch (right column) are shown for the ABO2 (top, blue symbols), WCO2 (middle, green symbols) and SCO2 (bottom, red symbols) channels. Both are shown on the left hand ordinate of each plot. The shifts are expressed in pixels, while the stretch is expressed as a ratio with the fixed dispersion stretch term, subtracted from 1, in parts per million. The shifts are well correlated with temperature gradients across the OBA (dark grey crosses and right hand ordinate), which can introduce grading tilts and other distortions in the OBA. The stretch term is correlated with the cryocooler temperature (dark grey symbols and right hand ordinate). These sifts appear to be correlated with time between Decon cycles, indicated as vertical grey bars.**



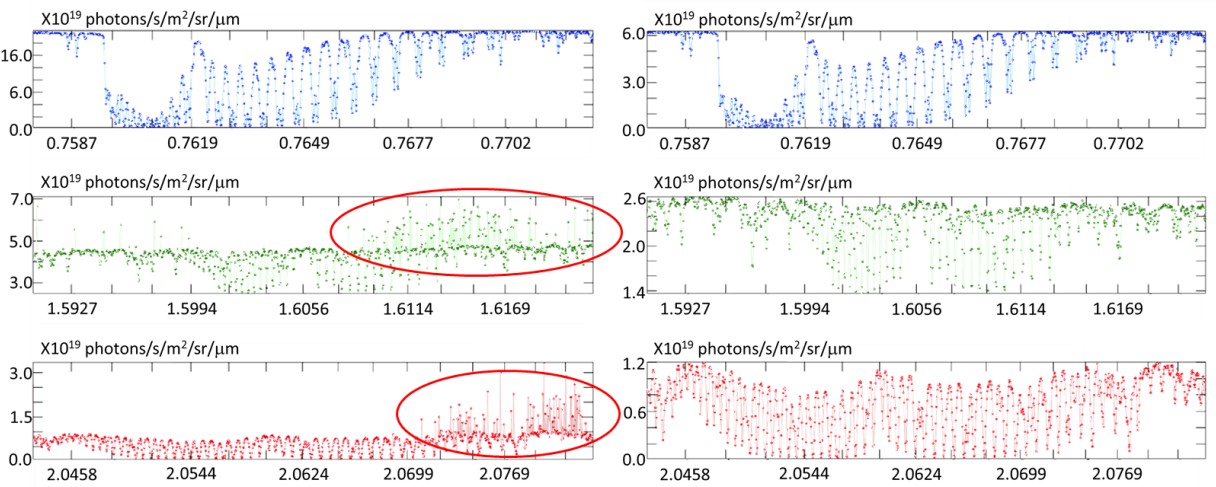

**Figure 8: Spectra recorded in footprint 3 of the ABO2 (top), WCO2 (middle) and SCO2 (bottom) channels on August 8 (left) and September 10 (right) show the impact of an improved dark calibration. The August 8 spectra were calibrated using FPA dark response recorded during the pre-launch thermo-vacuum tests at JPL in 2012. The September 10 spectra were calibrated using dark data recorded in flight. The revised dark data and bad pixel maps dramatically improve the quality of the spectra at longer wavelengths in the WCO2 and SCO2 channels.**





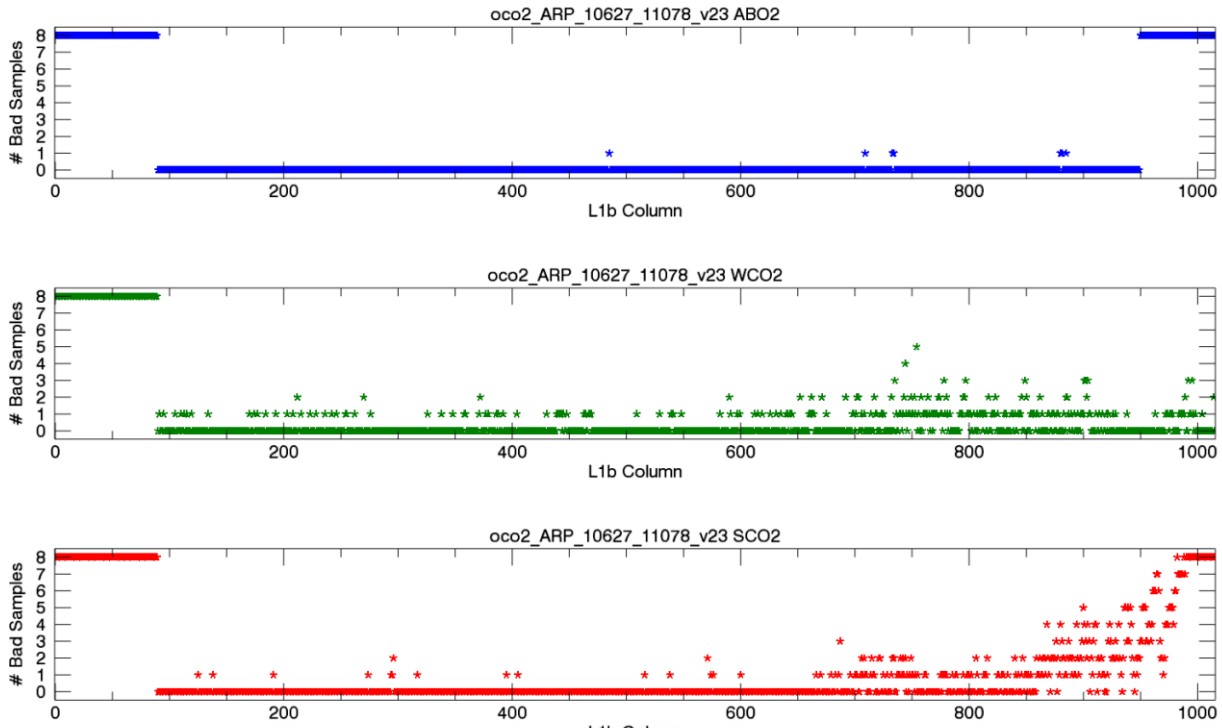

**Figure 9: The bad sample distribution is shown as a function of the L1B column number and footprint for the ABO2 (top), WCO2 (middle) and SCO2 (bottom) channels. Bad samples in footprints 1 through 8 are indicated by a \* in the corresponding y-axis value. All columns with all good samples are indicated by a star in y-axis value, 0. All samples in columns less than ~180 have been marked as bad on all 3 FPAs due to optical aberrations that are not well characterized on that end of the array. Samples near the right hand side of the FPAs have been marked as bad because the ILS is not well characterized there.**





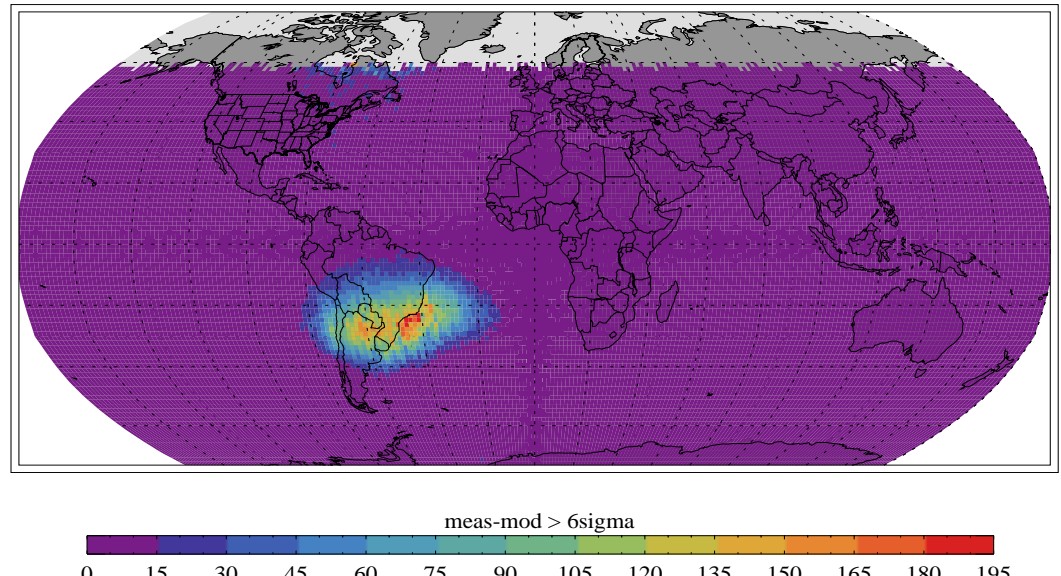

**Figure 10: The number of spectral samples in the O2 A band that are contaminated by 10-sigma cosmic ray events clearly shows the geographical extent of the South Atlantic Anomaly (SAA).**





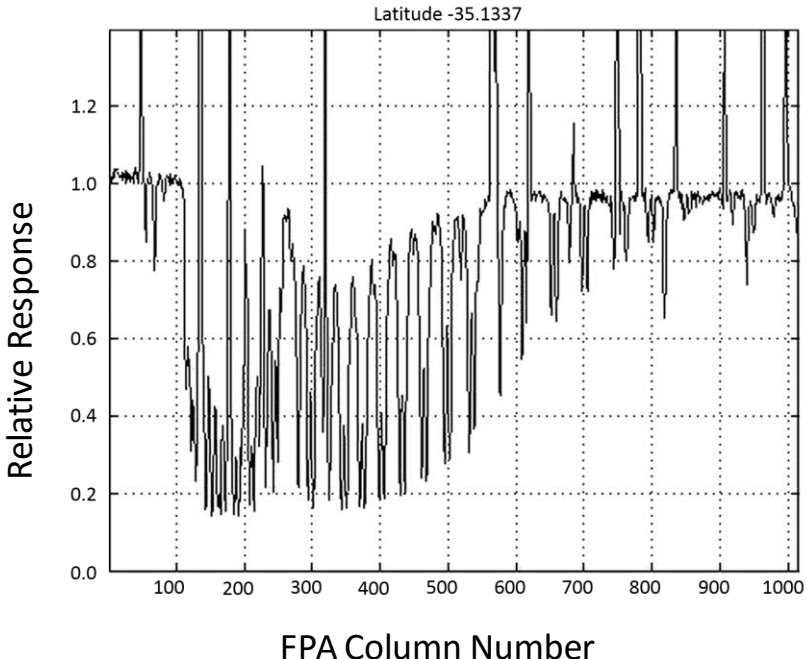

**Figure 11: A raw ABO2 spectrum showing the impact of a cosmic radiation event over the South Atlantic Anomaly. Cosmic rays produce spurious positive spikes that contaminate one or more adjacent pixels.**





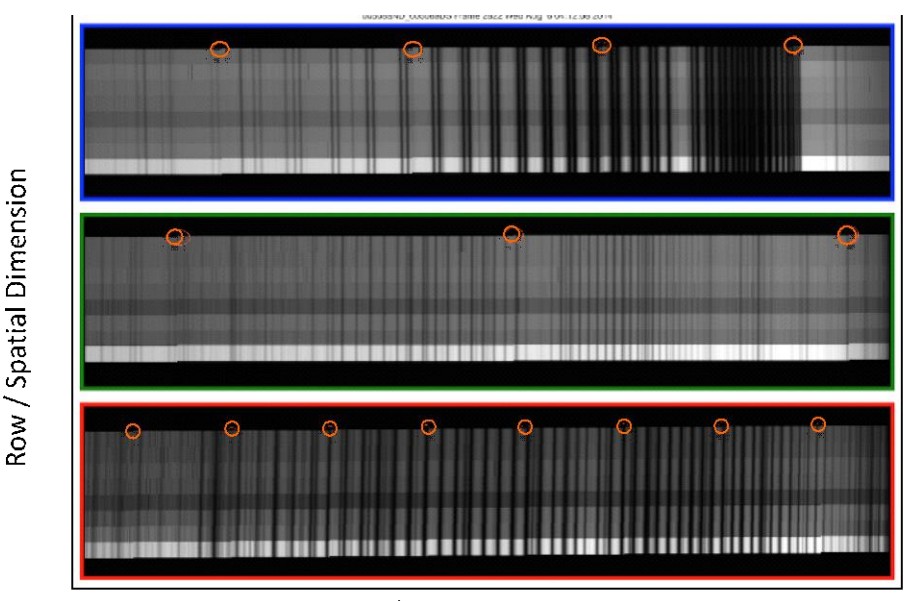

**Figure 12: Images of the ABO2 (top), WCO2 (middle) and SCO2 (bottom) channels in "summed mode" taken from the OCO-2 first light frame over Papua New Guinea. The 8 spatially-summed footprints in each band are shown from bottom (footprint 1) to top (footprint 8). In this 0.333 second frame, footprint 1 was contaminated by a cloud and appears brighter than the rest. This frame clearly shows the rotation, or "clocking" of each FPA columns with respect to the dark O2 and CO2 absorption lines. The O2 lines are tilted slightly counter-clockwise, while the CO2 lines in the WCO2 and SCO2 channels are tilted clockwise. The columns where the 1-pixel clocking adjustments are applied are highlighted with red circles.**





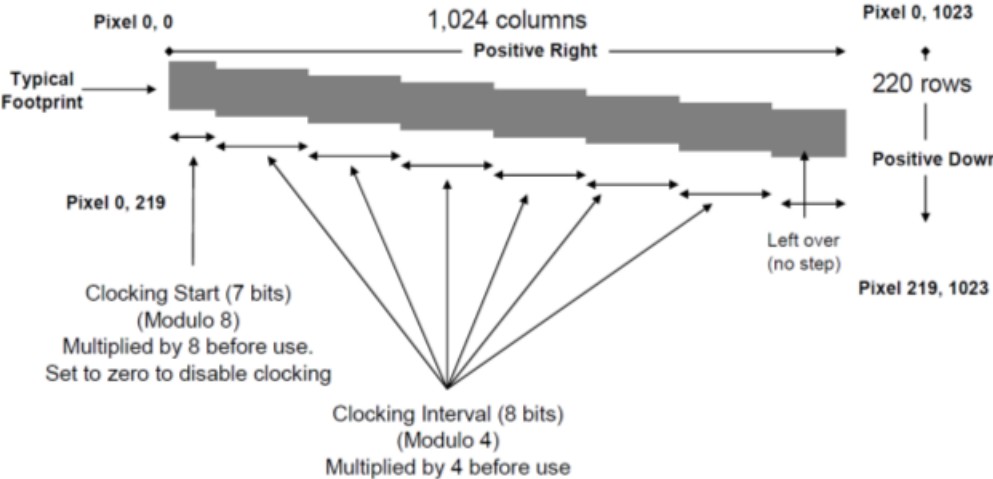

**Figure 13: The "clocking" scheme used to sample the same spatial information (vertical dimension) across the entire spectral range (horizontal) for a rotated FPA.**





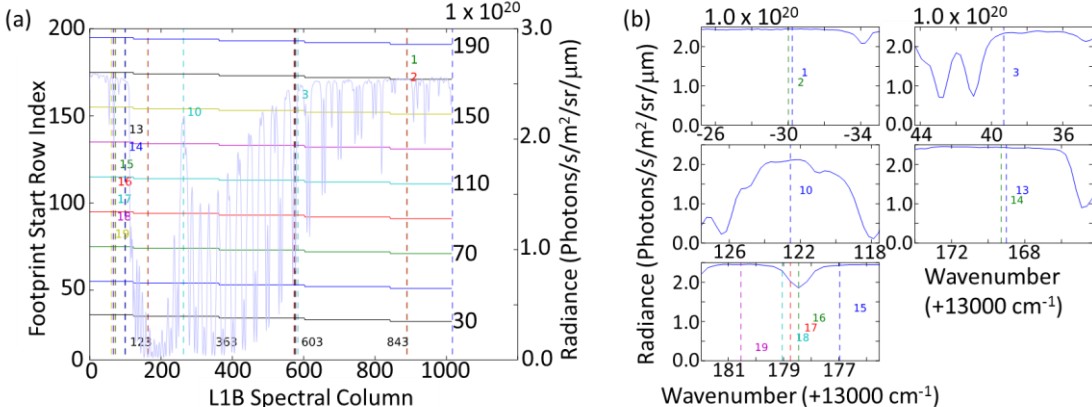

**Figure 14: (a) Schematic illustrating the clocking for the O₂ A-band along with the current color slice selection. Horizontal, piecewise-constant solid curves delineate footprint boundaries. The discrete jumps in the footprint boundaries produce the clocking discontinuities. Dashed vertical lines indicate color slice selection. Only those vertical lines that are numbered are used in the clocking correction algorithm. A typical A-Band spectrum is overlaid (blue line) for reference. (b) The color slices used in the clocking correction for the O2 A band are aggregated into five "groups." Each panel depicts a single group of color slices (dashed lines) used to identify radiance discontinuities. Spectral features for a "typical" O₂ A-band spectrum are shown as a solid blue line.**





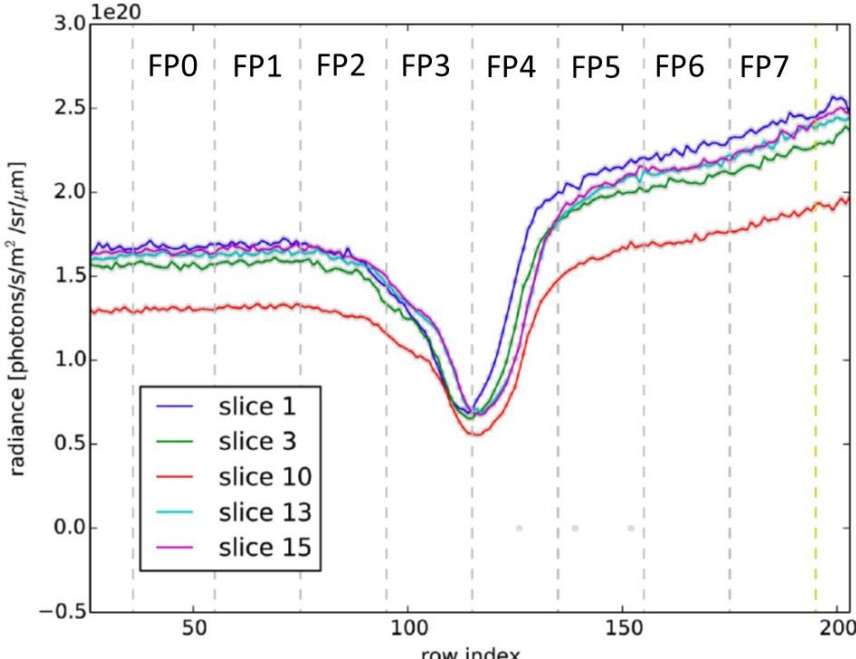

**Figure 15: O₂ A band radiances sampled for a few select color slices as a function of row index, which serves as a proxy for actual spatial variation within a scene. Solid curves are corrected values in which "bad" pixel data have been removed. Grey dashed vertical lines represent the boundaries of footprints (labeled FP0-FP7). The large differences between the values of the radiance on the boundaries of footprints 3 and 4 (starting from zero on the left) will produce large "clocking" spectral discontinuities in the measurement.**





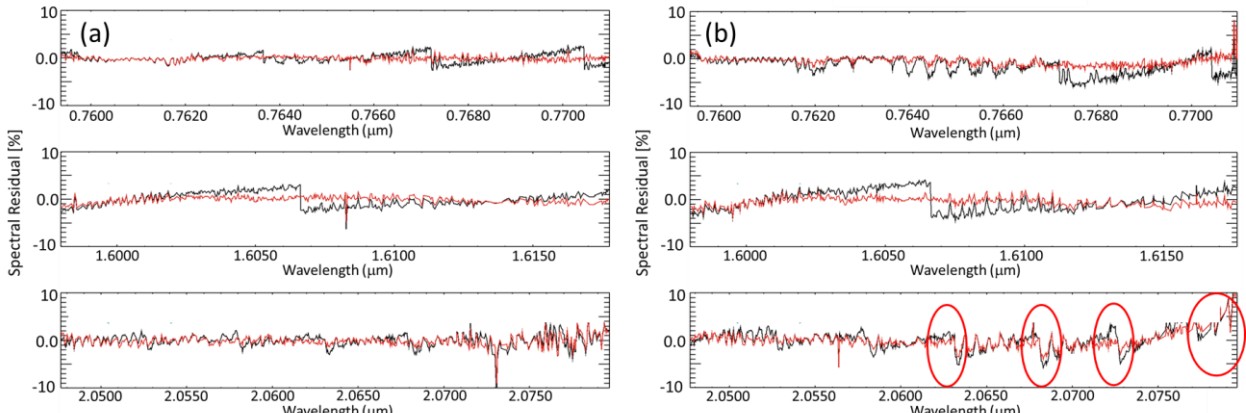

**Figure 16: The impact of clocking errors, and the results of the clocking correction algorithm are most easily seen in the spectrally dependent differences between the observed and simulated spectra. (a) Residuals for a spatially inhomogeneous scene are shown in black for the ABO2 (top), WCO2 (middle) and SCO2 (bottom) spectrometers. The corrected spectra are shown in red. Radiance discontinuities occur at the locations of single-pixel shifts in the start and stop row indices of the spectral sample. The amplitudes of the discontinuities are substantially reduced by the clocking correction algorithm. (b) Residuals for the ABO2 (top), WCO2 (middle) and SCO2 (bottom) are shown for a scene were the residuals are well corrected in the first two channels, but inadequately corrected in the SCO2 channel (red circles).**





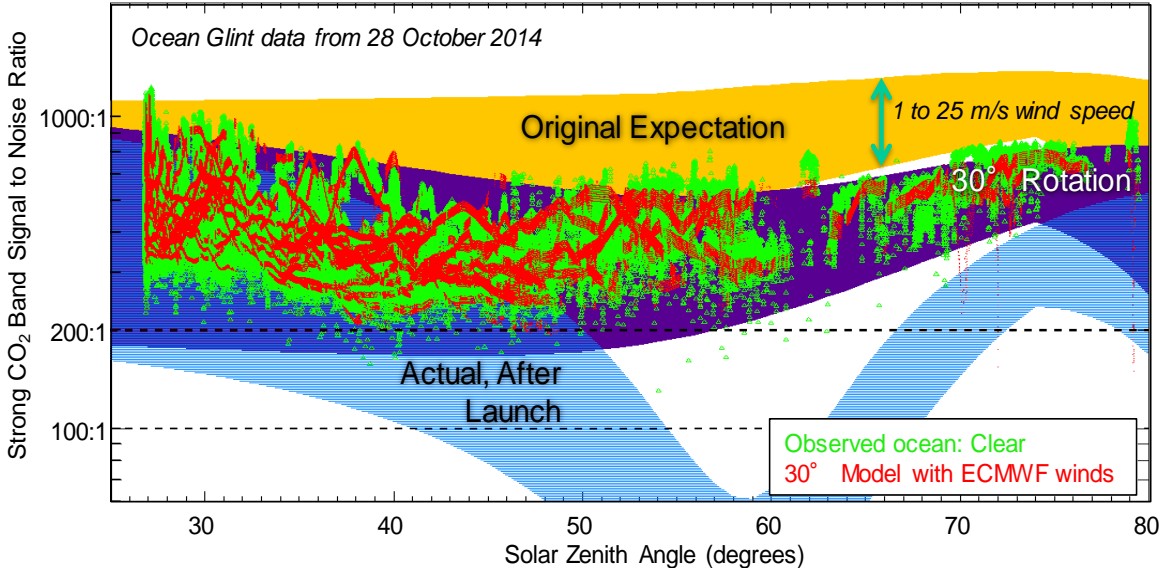

**Figure 17: The continuum signal to noise ratio (SNR) for ocean glint observations is shown as a function of solar zenith for the SCO2 channel. The gold band shows the performance originally predicted using a Cox-Munk ocean reflectance model for wind speeds varying from 1 to 25 m/sec. The actual performance observed immediately after launch, when the spectrometer slits were oriented perpendicular to the principal plane, fell within the light blue band. The low values between 50 and 60° are near the Brewster angle. Predictions from the Cox-Munk ocean reflectance model indicated that introducing a 30° yaw around the center of the instrument's field of view would yield values within the purple band (actual predictions shown as red points). After this change in the observing geometry was implemented, the observed SNR for clear-sky ocean soundings increased to values above 200:1 at most solar zenith angles (green points). The SNR in the other two channels is somewhat higher because the solar flux is larger there.**





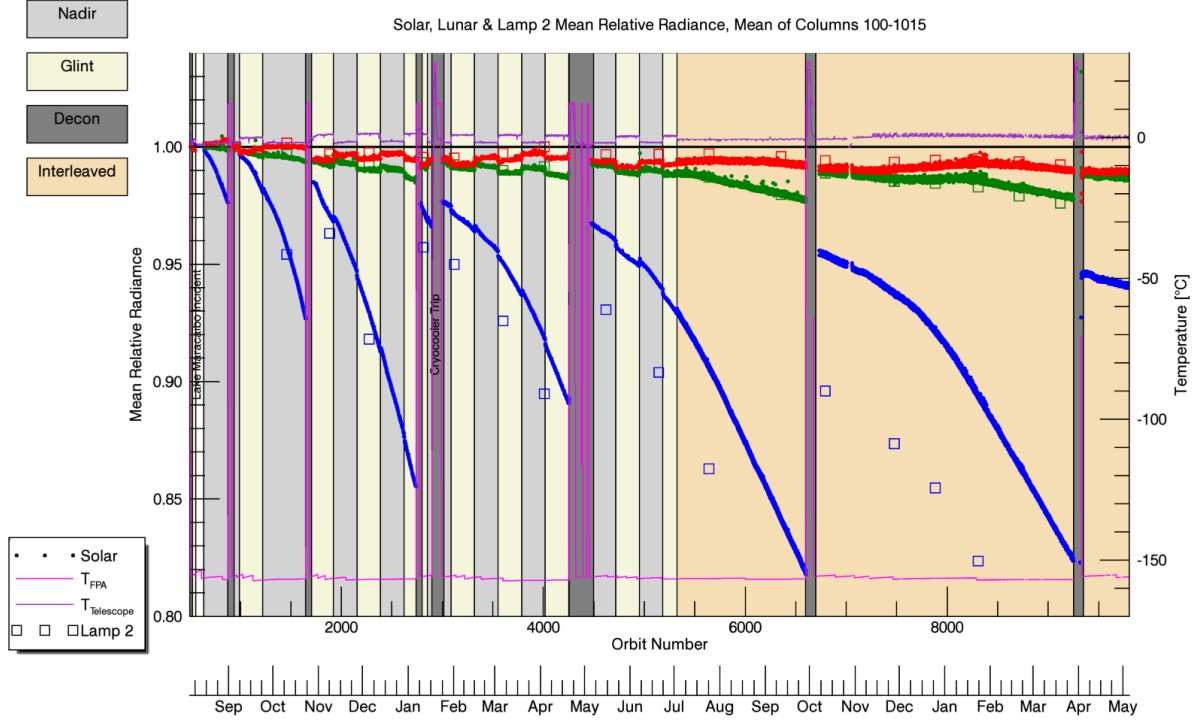

**Figure 18:** **The relative radiometric response of the ABO2 (blue), WCO2 (green) and SCO2 (red) channels is shown as a function of time (orbit number and month) for the OCO-2 instrument. All data are referenced the first on-orbit radiometric calibration data, collected in early August 2014. These results show the average response of all footprints in each channel. Routine solar diffuser observations are shown as points, while monthly observations collected with a backup lamp (Lamp 2) are shown as boxes. The instrument decontamination (decon) activities are indicated as dark grey boxes. The alternating 16-day glint and nadir observation periods performed during the first year of the mission are indicated as light yellow and light grey boxes. These observing modes modify the instrument and spacecraft thermal environment, and produced small, but measureable changes in the radiometric performance of the instrument and its calibration system. The FPA temperatures are indicated on the right hand axis. Early in the mission, decon activities were performed frequently to remove ice accumulation on the thermal straps connecting the FPAs to the cryocooler. These activities have become less frequent over time as the system has vented water into space.**





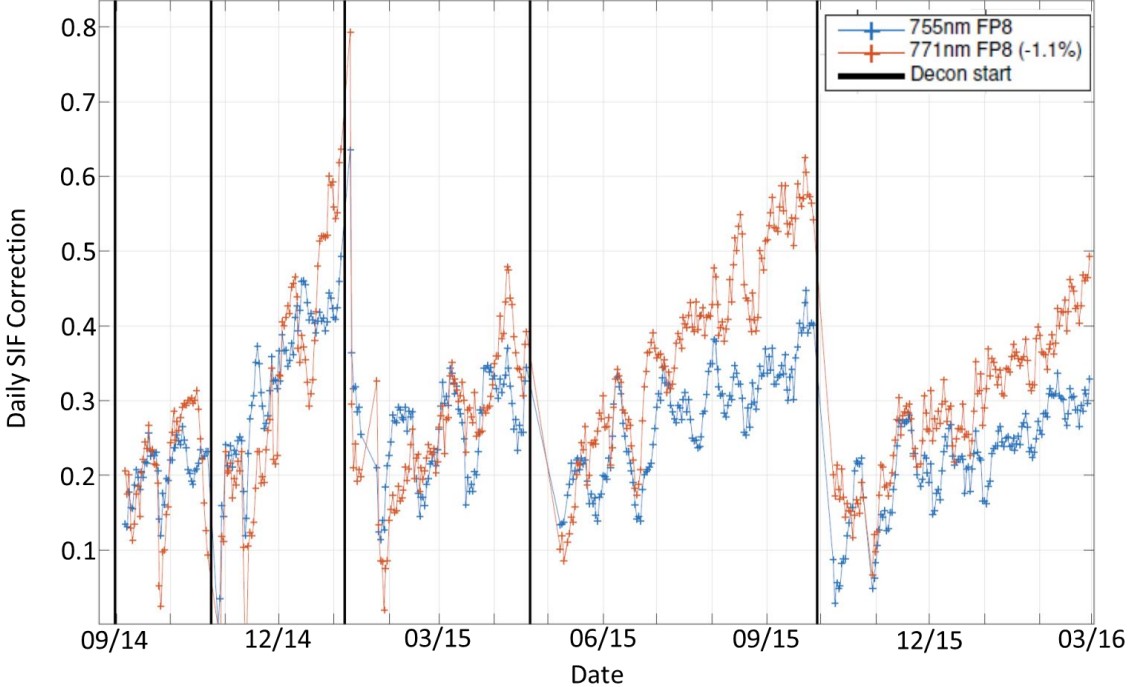

**Figure 19: The correction applied to the Solar Induced Chlorophyll Fluorescence (SIF) estimate is shown as a function of time for the solar lines located near 755 (blue) and 771 nm. The timing of the decon activities is indicated by the vertical black lines. These observations suggest that as ice accumulates on the ABO2 FPA, increasing its reflectance, a fraction of that light may be scattered back to the FPA by the cold filter, and other optics above the FPA, producing a zero level offset.**





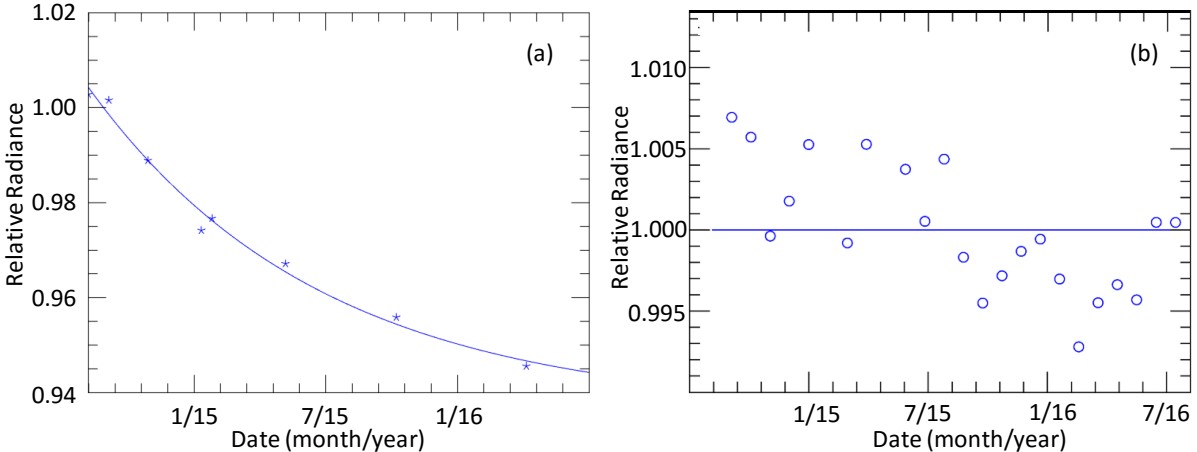

**Figure 20: (a) The "slow degradation" in the radiometric response immediately after decon activities (stars), as seen for observations of the Sun through the solar diffuser, can be fit by a smoothly varying function (solid line). The total degradation over the first 18 months in orbit is just over 5%. Approximately 80% of this degradation has been attributed to degradation of the solar diffuser plate itself, which is not in place during science observations. (b) Observations of the ~75% gibbous moon taken with the instrument aperture open (no diffuser) show a decrease in the relative response of the ABO2 channel that is less than 1% per year. These lunar calibration observations indicate that the remaining ~4% of slow degradation can be attributed to reductions in the gold coatings used on the solar diffuser.**