# Peer review of "The On-Orbit Performance of the Orbiting Carbon Observatory-2 (OCO-2) Instrument and its Radiometrically Calibrated Products"

_Atmospheric Measurement Techniques, 2016_

## Referee Comment (RC1) · Anonymous Referee #2 · 25 Oct 2016

Comments on The on-orbit performance of the Orbiting Carbon Observatory-2 (OCO-2) instrument and its radiometrically calibrated products by David Crisp et al

Overall comments: This paper describes how the radiometrically calibrated data products on-orbit from OCO-2 were determined. The paper is quite comprehensive and provides an excellent view of the calibration work the OCO-2 team has accomplished. Details are often not provided, but reference is made to a submitted paper by Rosenberg in which those details are presumably discussed. As expected, no science data is presented. Although some of the comments below are substantive, most of the comments are fairly trivial in nature. This paper is a valuable reference paper for use of the OCO-2 L1b and L2 data. Recommendation: accept subject to minor revisions as

noted below.

Specific comments:

general: - you should state what the absolute radiometric calibration goals were for OCO-2 and if those goals were met. You really only discuss relative calibrations and changes in the text. Details are likely in the Rosenberg paper, but a short paragraph would be valuable to set the stage for the reader

- you discuss briefly the ILS, perhaps expand that discussion and show a graph of the determined ILS for each channel since potential changes to the ILS are critical to "on-orbit performance of OCO-2".

specific:

1/21 "These" are particularly . . .. ; Remove "the"; spell "observatons" correctly

2/13 changes "in" the line core

15 the implication here is that there is a single spectrograph with 3 detectors in the focal plane, rather than 3 separate spectrographs. This is an important distinction due to scattered light considerations.

18 sensitivity add "(s/n > 400)"

20 remove terminology "full-physics". Say "detailed" or some such word if you want to emphasize its technical ability. Full physics is not very meaningful – can never be full physics

25 in the ( ), aren't these reversed relative to line 24?? The detectors should be coldest?

3/16 a "common" relay optics assembly

/18 perhaps add the level of rejection by the narrow pass filter

/21/22 the statement regarding the alignment of the 3 polarizers leaves open the pos-

sibility that the polarizer axes are not co-aligned since the spectrographs may have rotated polarization sensitivities. Please clarify.

/23 "a" spectrometer slit

/26 might want to mention level of thermal emission as a fraction of the continuum level in each channel?

/28 mention that only 160 pixels spatially are illuminated so the caption to figure 2a is more easily understood – only part of the spatial direction of the FPA is utilized

4/2 "returned as unilluminated reference pixels"

/11 "to" the FPAs . . ..

/19 might want to expand on the lack of need for a physical shutter. After readout, are the pixels reset to zero, or is there a potential memory effect due to residual charge

/24 replace "for" with "during" since this mode is only used occasionally

5/7 add to the list of mrad and deg, the projected size on the ground in km

6/12 really the measurement is of the absolute radiometric response of the instrument with any changes in the solar diffuser embedded in that measurement, as you note on line 21

9 the discussion at the top of pg 9 is confusing. 8/24 states dark offset of each pixel is sensitive to small (mK) changes in the temp of the FPA, whereas top of pg 9 states that the "dark offset is relatively insensitive to temperature". L3 states that a few samples have much greater temperature sensitivity (Fig 6). Fig 6 graphs are fairly clear, but what are the well-behaved and temp-sensitive samples? Why are some samples more sensitive than others? This is confusing.

/21 relative "to" the OBA

11/5 is it possible to quantify "measurable amount" in km projected at the surface of the

Earth?

/6 please give the indicated "specification" for alignment in km projected at the surface of the Earth

/8 add in parentheses the half size of the OCO-2 footprint in km

/30 which was – add space

/18 why are not gain corrections applied to individual pixels prior to incorporation? If because so small the weighting does not matter, state that.

12/26 does this mean that previously pixels labeled as bad were changed to be labeled as OK?

15/6 "v"7

16 you refer to figure 12 for the first time in L24, but before that refer to figs 13 and 14. Numbering of figs?

/24 might be interesting to give the angular rotation of each of the 3 channels – Figure 12 shows the angle to be nontrivial

17/21 do you mean screened "out"?

/32 "wavelength" dependent polarization . . ..

20/26 the "science" aperture is mentioned. Not entirely clear what this means. Perhaps change the sentence to read "These measurements are made in the normal Earth-observing mode without the solar or lamp diffusers in place and indicate . . ..."

21/4 how is it independently concluded that the lamps are decreasing in output since what is observed is the combination of lamp/diffuser that is actually observed?

/24 might want to note that the data products go back to October 2014 and not give the impression it is only from June 2015

22/10 might want to include a few words on bias correction

27 in Figure 2 caption, please give the spatial resolution of a super pixel

27 in Figure 2 you might want to note that columns are horizontal in this figure and rows are vertical. This is initially confusing while trying to digest this complicated figure since it is opposite of what one normally considers a column and a row.

30/ you might want to put horizontal lines on Fig 5 to show minimum required s/n for each channel for a single sounding

31/ L4,7 – the definition of "training range" is unclear here and on 9/2

32/ grating tilts not grading tilts?? Shifts not sifs. No label on RH ordinate (°C?) – looks cut off in pdf version?

33/ why are the bad pixels concentrated on the RHS?

34/ Fig 9 – last sentence, "right hand edge" as assume you mean the * that are just flat with L1b column?

37/ Fig 12 – might be nice to list the angular tilt of each band

39/ Fig 14 (b) – might want to comment on why columns around 179 wavenumbers are not chosen in a flat continuum region??

40/ tell the reader why the SCO2 channel is not properly corrected (cloud?) – not explained in text 17/20

42/ Fig 17 – add to the caption 18/23 - Note that SNR values near 200 are needed to yield XCO2 estimates with single sounding random errors less than 1 ppm.

43/ Fig 18 L2 – all data are referenced "to" the . . ..

the figure is labeled "solar, lunar, and lamp 2 mean . . ...", but in the symbol label on the lower LHS, there is no symbol for lunar data, nor is lunar mentioned in the caption; it is discussed in the text

---

## Referee Comment (RC2) · Anonymous Referee #3 · 30 Oct 2016

This paper describes a latest performance of the OCO-2 radiance data products. The paper is well written and the topic is suitable for AMT. I recommend it to be published after the following comments are addressed.

p.2 l.30 : The order of the operating temperatures in the parentheses is reversed.

p.13 l.17 : According to Table 3, a single bad sample is added on "1" July 2015 in the "WCO2" channel.

p.15 l.10 : The "V"7 cloud screening

p.16 l.24 : According to Table 4 and Figure 12, there are "3" discontinuities in the WCO2 channel.

[Figure]

p.18 l.32 : (Figure "17")

p.24 l.7 : Haring et al. 2002 is not referred in the text.

p.29 Figure 4 : It is better to unify the temperature unit in °C.

p.44 l.2 : 755 (blue) and 771 nm "(red)"

———————————————

---

## Referee Comment (RC3) · Anonymous Referee #4 · 30 Oct 2016

<General Comments> It was very challenging to monitor carbon dioxide from space using imaging grating spectrometer technology. Accurate and precise XCO2 retrieval need radiometric, spectroscopic, geometric, and polarimetric calibrations and characterizations. The history to investigate, mitigate, correct and refine anomalies after launch is very important. Some additional information below will improve readers' understanding. The satellite operation might not be a topic of the paper but brief explanation on target observation selection, priority, and frequency will help readers' understanding. An idea to minimize the focal plane assembly rotation by design or operation, discussion on necessity of onboard calibration using a solar diffuser for future missions such as OCO-3 will benefit satellite GHG remote sensing community. Onboard so-

lar differs on spaceborne optical instruments often have larger degradation than the telescope and spectrometers. I recommend publication after minor revision.

<Specific Comments> (1) Page 2, Lines 29-30, "The optical bench and focal planes …... operating temperatures (near -152.4 C and -6.4 °C, respectively" Is it near and -6.4 °C and -152.4 C, respectively?

(2) Page 11, Lines 17-18, "the relative radiometric performance (zero level offset, gain, and gain linearity) of spectral samples within a given channel must be known to within 0.1%" What is the definition of zero level offset, gain, and gain linearity of 0.1%?

(3) Page 19, Line 18, "a thick layer of ice would significantly alter the ABO2 instrument line shape (ILS) function," Is the reason why thick layer affects ILS the mechanism described in page 10, lines 13 – 20?

(4) Page 30, Figure 5. (a) Definition of "maximum measurable signal" will be helpful for readers. Is it dynamic range of the AD converter?ãĂĂDo the data used in Figure 5 (a) include cloud contaminated scene? How do authors calculate SNR from observed data?

(5) Page 35, Figure 10. Which period of data is used? For 18 months?

(8) Page 43, Figure 18 "with a backup lamp (Lamp 2)" Why back up lamp data is used in calibration? How to use and compare primary and back up (monthly) lamps on orbit? How about the third one?

<Technical Corrections> (1) Page 13 Table 3 What does "BPM" stand for?ãĂĂãĂĂIs it "bad pixel map"?

(2) Page 15, Line 10, The 7 > The V7?

(3) Figures 10, 12 and 14, Captions O2A, O2, CO2, O2A: "2"s are subscript.

(4) Page 42, Figure 17, Caption, Green points Are they "green triangles?"

---

## Author Comment (AC1) · 6 Nov 2016

We greatly appreciate the detailed review by Anonymous Referee #2. We have attempted to address all of the issues that they have raised. These changes are documented here and in the text. We hope these changes adequately address these concerns.

Referee # 2: general: - you should state what the absolute radiometric calibration goals were for OCO-2 and if those goals were met. You really only discuss relative calibrations and changes in the text. Details are likely in the Rosenberg paper, but a short paragraph would be valuable to set the stage for the reader.

A brief summary of the absolute radiometric calibration was added to section 5.1, stating: Prior to launch, the absolute radiometric calibration of the instrument was established from observation of an integrating sphere, with reference radiometers validated against National Institute of Standards and Technology (NIST). This system was required to meet a 5% absolute radiometric requirement, but yielded much smaller uncertainties (1.6%, 3.2%, and 2.9% in the ABO2, WCO2, and SCO2 channels, respectively; Rosenberg et al., 2016).

- you discuss briefly the ILS, perhaps expand that discussion and show a graph of the determined ILS for each channel since potential changes to the ILS are critical to "on-orbit performance of OCO-2".

The ILS was not discussed in detail here because it is the primary topic of the papers by Frankenberg et al. 2014 and Lee et al. (2016), which describe its pre-launch characterization and calibration. At this point, there is no evidence that the ILS has changed by a measureable amount, as noted in section 5.2. This conclusion is reinforced by a the paper by Sun et al. that has recently be submitted to this journal. It is not clear that we can add anything to the previous papers, without simply repeating their results verbatim.

specific: 1/21 "These" are particularly : : :. ; Remove "the"; spell "observatons" correctly Reworded as "This is a particularly challenging remote sensing observation because all but the largest ..."

2/13 changes "in" the line core done

15 the implication here is that there is a single spectrograph with 3 detectors in the focal plane, rather than 3 separate spectrographs. This is an important distinction due to scattered light considerations.

Reworded as: "To record these small changes in the reflected solar spectrum, OCO-2 carries and points a single instrument that incorporates 3 imaging grating spectrometer

channels ..."

18 sensitivity add "(s/n > 400)"

Added: "(continuum signal-to-noise ratio typically > 400)"

20 remove terminology "full-physics". Say "detailed" or some such word if you want to emphasize its technical ability. Full physics is not very meaningful – can never be full physics

Changed to "Coincident measurements from the three spectral channels are combined into "soundings" that are analyzed with a state-of-the-art retrieval algorithm"

25 in the ( ), aren't these reversed relative to line 24?? The detectors should be coldest?

Corrected. It now reads: ". . . cooled to their operating temperatures (near -6.4 °C and -152.4 ïĆřC, respectively)"

3/16 a "common" relay optics assembly Added.

/18 perhaps add the level of rejection by the narrow pass filter

It varies, but we added "(out-of-band transmission < 10-4 of peak transmission).

/21/22 the statement regarding the alignment of the 3 polarizers leaves open the possibility that the polarizer axes are not co-aligned since the spectrographs may have rotated polarization sensitivities. Please clarify.

This was modified to "light light polarized perpendicular to the long axis of the slits (e.g. in the direction of dispersion)"

/23 "a" spectrometer slit

Done.

/26 might want to mention level of thermal emission as a fraction of the continuum level

in each channel?

We do not discuss thermal emission in detail here because it is negligible in the ABO2 channel, and is reduced to near zero counts in the WCO2 channel. It produces a small offset in the SCO2 channel that is removed as part of the dark offset calibration, introducing negligible noise (< 15 counts, which is less than half of the detector read noise). We did add the following clarification to the description of the cold filter at the bottom of pg 3: "A second, narrowband filter, which is cooled to approximately -93 ïĊřC , has been installed just above each FPA to further reduce the out-of-band light at wavelengths > 2% away from the central wavelength of the channel. This filter also limits the impact of thermal emission from the optical bench, which would otherwise introduce a source of noise in the CO2 channels."

/28 mention that only 160 pixels spatially are illuminated so the caption to figure 2a is more easily understood – only part of the spatial direction of the FPA is utilized

Added statement "The slits illuminiate only the central ∼190 of the 1024 pixels on each FPA (Figure 2a)."

4/2 "returned as unilluminated reference pixels"

Added

/11 "to" the FPAs . . .

Corrected.

/19 might want to expand on the lack of need for a physical shutter. After readout, are the pixels reset to zero, or is there a potential memory effect due to residual charge

Modified: "For routine science operations, a 220 (spatial rows) by 1016 (spectral columns) pixel window on each FPA is continuously scanned using a "rolling readout" method for recording and resetting each pixel on the FPAs to their bias levels (Haring et al. 2004)."

/24 replace "for" with "during" since this mode is only used occasionally

Done.

5/7 add to the list of mrad and deg, the projected size on the ground in km

Done. "0.14 mrad (∼0.1 km at nadir),"

6/12 really the measurement is of the absolute radiometric response of the instrument with any changes in the solar diffuser embedded in that measurement, as you note on line 21

Added parenthetic reference to solar calibrator: "(and solar calibrator)"

9 the discussion at the top of pg 9 is confusing. 8/24 states dark offset of each pixel is sensitive to small (mK) changes in the temp of the FPA, whereas top of pg 9 states that the "dark offset is relatively insensitive to temperature". L3 states that a few samples have much greater temperature sensitivity (Fig 6). Fig 6 graphs are fairly clear, but what are the well-behaved and temp-sensitive samples? Why are some samples more sensitive than others? This is confusing.

We clarified the statement on 8/24 to read ". This component of response must be updated frequently in orbit because the dark offset of a few pixels is sensitive to small (millikelvin, mK) changes in the temperature of the FPA." The discussion that follows now states "For most spectral samples, this extrapolation has minimal impact because the dark offset is relatively insensitive to temperature, while a few samples have much greater temperature sensitivity (Figure 6)."

/21 relative "to" the OBA

Done.

11/5 is it possible to quantify "measurable amount" in km projected at the surface of the Earth?

Modified to "measurable amount (∼0.07 mrad, or 50 m at nadir)"

/6 please give the indicated "specification" for alignment in km projected at the surface of the Earth

Modified to: "was within specification (< 1ïĆř)"

/8 add in parentheses the half size of the OCO-2 footprint in km

Modified to quantify actual detection limits: "This effort yielded geolocation errors no larger than 0.25 mrad (0.2 km at nadir), which is much smaller than the specification (0.9 mrad per axis, 3ïĄş or ∼0.9 km worst case at nadir). "

/30 which was – add space

Done.

/18 why are not gain corrections applied to individual pixels prior to incorporation? If because so small the weighting does not matter, state that.

To clarify this, we added the phrase "sample because the instrument controller was not fast enough to perform this calculation on board."

12/26 does this mean that previously pixels labeled as bad were changed to be labeled as OK?

To clarify, we added the parenthetic comment: (Note, while bad samples can be recovered through further calibration, individual pixels labeled as "bad" are not subsequently relabeled as "good").

15/6 "v"7

Done

16 you refer to figure 12 for the first time in L24, but before that refer to figs 13 and 14. Numbering of figs?

[Figure]

A reference to Figure 12 was omitted by accident in the first paragraph of this sub-section. It has been included in the sentence: "The focal plane arrays are therefore slightly rotated, or "clocked," with respect to the slit and grating (Figure 12)."

/24 might be interesting to give the angular rotation of each of the 3 channels – Figure 12 shows the angle to be nontrivial

The rotation angles are now included: " ... small ($\sim$0.3ïĆř counter-clockwise, 0.2ïĆř clockwise, and 0.5ïĆř clockwise, for the ABO2, WCO2, and SCO2, respectively),"

17/21 do you mean screened "out"?

Yes. We made that change.

/32 "wavelength" dependent polarization ...

Changed.

20/26 the "science" aperture is mentioned. Not entirely clear what this means. Perhaps change the sentence to read "These measurements are made in the normal Earth observing mode without the solar or lamp diffusers in place and indicate ..."

We modified this to "These measurements, which are made in the normal Earth-observing mode, without the diffuser in place, indicate that only about one-fifth of the observed attenuation can be attributed to reductions in the throughput of the telescope and spectrometers."

21/4 how is it independently concluded that the lamps are decreasing in output since what is observed is the combination of lamp/diffuser that is actually observed?

We clarified this as follows: "Comparisons of results obtained using the primary cali-bration lamp, which is used on all nominal polar calibration orbits (Lamp 1) to that of the other two lamps, which are used less frequently (Lamps 2 and 3) indicate that the output of this calibration lamp has also decreased somewhat in the ABO2 channel."

[Figure]

/24 might want to note that the data products go back to October 2014 and not give the impression it is only from June 2015

We made the time range more explicit: "Starting in June of 2015, the OCO-2 team began reprocessing the entire OCO-2 data record, extending back to September 6, 2014, using the V7/7r algorithm and delivering this product to the GES-DISC for distribution to the science community."

22/10 might want to include a few words on bias correction

This is a very long topic that could substantially increase the length of the paper. It is also covered well by Eldering et al. (2016) and in a far more detailed paper that is currently under preparation (Odell et al.). Here, we directed the reader to Eldering et al. by adding the statement: "The impact of these uncertainties on the L2 products are being evaluated using comparisons with observations from the TCCON network and other standards. Using these comparisons, a bias correction has been developed and delivered to the comunity in the V7 "Lite Files" (Eldering et al. 2016)."

27 in Figure 2 caption, please give the spatial resolution of a super pixel

We added the sentence: "Each footprint has a cross-track dimension of < 1.3 km and a down-track dimension of $\sim$2.3 km at nadir"

27 in Figure 2 you might want to note that columns are horizontal in this figure and rows are vertical.

This is initially confusing while trying to digest this complicated figure since it is opposite of what one normally considers a column and a row. We modified the figure to avoid this potential source of confusion.

30/ you might want to put horizontal lines on Fig 5 to show minimum required s/n for each channel for a single sounding

The minimum continuum SNR needed for a single sounding is difficult to quantify exnone

actly, but something around 200 usually yields XCO2 estimates with single sounding random errors near 1 ppm. We have added this note the caption: "A continuum SNR exceeding 200 is typically needed to yield single sounding random errors near 1 ppm."

31/ L4,7 – the definition of "training range" is unclear here and on 9/2

We modified the wording as follows: ". If the FPA or OBA temperatures move outside the range of values used in these fits"

32/ grating tilts not grading tilts?? Shifts not sifs. No label on RH ordinate (C?) – looks cut off in pdf version?

The typos were corrected and the plot size was reduced to avoid truncating the RH axes labels.

33/ why are the bad pixels concentrated on the RHS?

The short answer is "we don't know." As noted in the text, these are very old FPAs. Bad pixels often cluster in discrete areas of hybridized devices, possibly reflecting manufacturing issues or stresses encountered during storage or use. To address this question, we added sentence to the caption of Figure 9: "A larger number of bad samples are marked on the right hand side of the WCO2 and SCO2 FPAs because more bad pixels have appeared on that side of these FPAs."

34/ Fig 9 – last sentence, "right hand edge" as assume you mean the * that are just flat with L1b column?

No. The ILS was difficult to quantify the ILS accurately for 20-50 columns near both edges of the FPAs. The quality of the calibration on the LHS is compromised by optical aberrations, while the quality of the ILS calibration near the RHS is compromised only by limitations our ability to fully sample both sides of the ILS for those pixels. The ILS calibration is described further in Lee et al. 2016.

37/ Fig 12 – might be nice to list the angular tilt of each band

We have included this in the text and in the figure caption by adding the sentence "The O2 lines are tilted counter-clockwise by 0.3ïČř, while the CO2 lines in the WCO2 and SCO2 channels are tilted clockwise by ∼0.2ïČř and 0.5ïČř, respectively."

39/ Fig 14 (b) – might want to comment on why columns around 179 wavenumbers are not chosen in a flat continuum region??

We added a sentence to the caption stating "Other color slices in strongly absorbing regions (i.e. those near columns 200 and 600) were intended for cloud screening applications. "

40/ tell the reader why the SCO2 channel is not properly corrected (cloud?) – not explained in text 17/20

We attempted to clarify this by adding the sentence: "The cause for the poor fit in the SCO2 channel is unknown, but may be related to the lack of true continuum in the SCO2 channel."

42/ Fig 17 – add to the caption 18/23 - Note that SNR values near 200 are needed to yield XCO2 estimates with single sounding random errors less than 1 ppm.

We added the statement "This is adequate to yield XCO2 estimates with single-sounding random errors near 1.0 ppm.

43/ Fig 18 L2 – all data are referenced "to" the . . . the figure is labeled "solar, lunar, and lamp 2 mean : : :..", but in the symbol label on the lower LHS, there is no symbol for lunar data, nor is lunar mentioned in the caption; it is discussed in the text

The typo in the caption was corrected. The title was also corrected, since the lunar data were removed from this plot for simplicity.

---

## Author Comment (AC2) · 6 Nov 2016

We greatly appreciate the review by anonymous referee #3. We have attempted to address all of the issues that they have raised. These changes are documented here and in the text. We hope this adequately addresses this concerns.

p.2 l.30 : The order of the operating temperatures in the parentheses is reversed.

Corrected. It now reads: ". . . cooled to their operating temperatures (near -6.4 °C and -152.4 ïĆřC, respectively)"

p.13 l.17 : According to Table 3, a single bad sample is added on "1" July 2015 in the "WCO2" channel.

[Figure]

Corrected in text. It now reads "was increased by one sample on 1 July 2015"

p.15 l.10 : The "V"7 cloud screening

Typo corrected. It now reads "The V7 cloud screening . . ."

p.16 l.24 : According to Table 4 and Figure 12, there are "3" discontinuities in the WCO2 channel.

Corrected. It now reads: "to only 3 in the WCO2 channel"

p.18 l.32 : (Figure "17")

Corrected.

p.24 l.7 : Haring et al. 2002 is not referred in the text.

This reference has been removed from the list.

p.29 Figure 4 : It is better to unify the temperature unit in _C.

We find that when we label both axes in degrees C, readers often believe that they can exchange the axes. This is not the case, since the right-hand axis for the cold head (CH) has been stretched and shifted relative to the on right for the FPAs. To clarify this, we have added the note: " The CH temperatures are shown as grey points (horizontal grey lines) in the top plot, and their values are labeled on the right hand y-axis (note: CH temperatures are shifted and stretched relative to the FPA temperatures and expressed in Kelvin rather than ïĆřC to avoid confusion with the FPA temperatures)."

p.44 l.2 : 755 (blue) and 771 nm "(red)"

Done.

---

## Author Comment (AC3) · 6 Nov 2016

We greatly appreciate the detailed review by anonymous referee #4. We have attempted to address all of the issues that they have raised. These changes are documented here and in the text. We hope these changes adequately address these concerns.

<Specific Comments>

(1) Page 2, Lines 29-30, "The optical bench and focal planes : : :.. operating temperatures (near -152.4 C and -6.4 _C, respectively" Is it near and -6.4 _C and -152.4 C, respectively?

[Figure]

Corrected. It now reads: ". . . cooled to their operating temperatures (near -6.4 °C and -152.4 ïČřC, respectively)"

(2) Page 11, Lines 17-18, "the relative radiometric performance (zero level offset, gain, and gain linearity) of spectral samples within a given channel must be known to within 0.1%" What is the definition of zero level offset, gain, and gain linearity of 0.1%?

Here, we changed the term "zero level offset" to " dark offset" to make this more consistent with the discussion of bad pixels. The gain and gain linearity are now explicitly defined in terms of the continuum radiance level: " For example, the relative radiometric performance (defined in terms of the dark offset, gain, and gain linearity; see Rosenberg et al. 2016) of spectral samples within a given channel must be known to within 0.1% of the continuum brightness across the spectral range of each channel to fully exploit the spectrally-dependent information in each sounding."

(3) Page 19, Line 18, "a thick layer of ice would significantly alter the ABO2 instrument line shape (ILS) function," Is the reason why thick layer affects ILS the mechanism described in page 10, lines 13 – 20?

No. The dispersion (not ILS) changes discussed on pg 10, lines 13-20 are not produced by a thick layer of ice on the FPAs. Those changes in dispersion (not ILS) are caused with physical distortions in the optical bench and FPA mounts associated with thermal stresses imposed by ice accumulation on the thermal straps that connect the cryocooler and FPAs. Thick layers of ice on the FPAs would produce ice lenses, that would distort the light incident on the FPAs. There is no evidence of that effect on OCO-2.

(4) Page 30, Figure 5. (a) Definition of "maximum measurable signal" will be helpful for readers. Is it dynamic range of the AD converter? Do the data used in Figure 5 (a) include cloud contaminated scene? How do authors calculate SNR from observed data?

[Figure]

We now explicitly define "maximum measurable signal" in the text with the parenthetic note: "(i.e. the maximum signal for which the instrument meets its radiometric calibration standards, see Table 1 and Rosenberg et al., 2016)." The data shown in the maps in Figure 5b have been screened for optically-thick clouds. The SNR calculation is described in detail in the L1B ATBD (Eldering et al. 2016a) and in greater detail in Rosenberg et al. (2016). Repeating that description once again here would add substantially to the length of this paper. We did add a statement near the top of section 5.1 stating "(see Eldering et al.; 2015 and Rosenberg et al. 2016 for a description of the methods used to derive the SNR).

(5) Page 35, Figure 10. Which period of data is used? For 18 months?

This figure was generated using data collected between November 2014 and January 2015, as now noted in the figure caption.

(8) Page 43, Figure 18 "with a backup lamp (Lamp 2)" Why back up lamp data is used in calibration? How to use and compare primary and back up (monthly) lamps on orbit? How about the third one?

To clarify this point, we have added the following sentence to the description of the calibration channel in Section 3: ". Lamp 1 is used for routine calibration (every orbit not used for downlink), while lamps 2 and 3 are used less frequently (monthly) to track degradation in lamp 1." Also, in section 6.5.2, we note "Comparisons of results obtained using the primary calibration lamp, which is used on all nominal polar calibration orbits (Lamp 1) to that of the other two lamps, which are used less frequently (Lamps 2 and 3) indicate that the output of this calibration lamp has also decreased somewhat in the ABO2 channel."

We use lamp 2 in this figure to reduce uncertainties due to the ∼1% lamp 1 degradation.

<Technical Corrections>

(1) Page 13 Table 3 What does "BPM" stand for? Is it "bad pixel map"?

Yes, we made that change in Table 3.

(2) Page 15, Line 10, The 7 > The V7?

Yes. We made that change.

(3) Figures 10, 12 and 14, Captions O2A, O2, CO2, O2A: "2"s are subscript.

Done.

(4) Page 42, Figure 17, Caption, Green points Are they "green triangles?"

Yes, we made that change in the caption.